# Light-induced primary amines and o-nitrobenzyl alcohols cyclization as a versatile photoclick reaction for modular conjugation

An-Di Guo[1,2,4], Dan Wei[1,2,4], Hui-Jun Nie[1,4], Hao Hu [3], Chengyuan Peng[3], Shao-Tong Li[1,2], Ke-Nian Yan[1,2], Bin-Shan Zhou[1], Lei Feng[1,2], Chao Fang[1], Minjia Tan [3], Ruimin Huang [3] & Xiao-Hua Chen [1,2✉]

The advent of click chemistry has had a profound impact on many fields and fueled a need for reliable reactions to expand the click chemistry toolkit. However, developing new systems to fulfill the click chemistry criteria remains highly desirable yet challenging. Here, we report the development of light-induced primary amines and o-nitrobenzyl alcohols cyclization (PANAC) as a photoclick reaction via primary amines as direct click handle, to rapid and modular functionalization of diverse small molecules and native biomolecules. With intrinsic advantages of temporal control, good biocompatibility, reliable chemoselectivity, excellent efficiency, readily accessible reactants, operational simplicity and mild conditions, the PANAC photoclick is robust for direct diversification of pharmaceuticals and biorelevant molecules, lysine-specific modifications of unprotected peptides and native proteins in vitro, temporal profiling of endogenous kinases and organelle-targeted labeling in living systems. This strategy provides a versatile platform for organic synthesis, bioconjugation, medicinal chemistry, chemical biology and materials science.

[1] Chinese Academy of Sciences Key Laboratory of Receptor Research, Shanghai Institute of Materia Medica, Chinese Academy of Sciences, Shanghai 201203, China. [2] University of Chinese Academy of Sciences, No. 19A Yuquan Road, Beijing 100049, China. [3] State Key Laboratory of Drug Research, Shanghai Institute of Materia Medica, Chinese Academy of Sciences, Shanghai 201203, China. [4] These authors contributed equally: An-Di Guo, Dan Wei, Hui-Jun Nie. ✉email: xhchen@simm.ac.cn

Inspired by nature's utilization of simple and powerful connecting reactions, the concept of click chemistry was first introduced by Kolb, Finn, and Sharpless as a synthetic strategy in 2001 (ref. [1]). Currently, click chemistry enabling rapid access to modular synthesis and bioconjugation, has become one of the most robust molecular assembly strategies in synthetic chemistry, modern drug discovery, biological research, nanotechnology, and materials science[2–9], providing remarkable alternatives to conventional chemistry[10–14]. Generally, click chemistry refers to a class of reactions that satisfy certain characteristics, such as modularity, operational simplicity (e.g., be insensitive to oxygen or water), reliable selectivity and high yields[1]. In addition, the ideal reactants for click chemistry should be easily accessible and diverse[1,7]. Over the past decade, considerable efforts have been devoted toward developing diverse chemical transformations as click reactions[15–17]. Notably, bioorthogonal click reactions have merged as highly specific tools based on genetic, metabolic or chemical incorporation of exogenous click handles to investigate the dynamics and function of biomolecules in vitro or in living systems[9,18]. Indeed, the performance of bioorthogonal click chemistries is greatly depending on the incorporation efficiency and the stability of the exogenous click handles in biological environments, for example, certain introduced functionalities may undergo side reactions with cellular nucleophiles[17]. On the other hand, the development of reliable chemoselective reactions with click characteristics for introducing functional motifs (FM) into the natural amino acid residues of proteins and peptides has appeared recently[9], such as 2-cyanobenzothiazole and N-terminal cysteine click reaction[19–21], tyrosine click reaction[22,23], amine bioconjugations[9,24–26], and sulfur (VI) fluoride exchange reaction[7]. However, the spontaneous manner of these kinds of click reactions for native biomolecules remains challenging when applied into complex biological environments, since the reactions would initiate in extracellular environment or during the process approaching cellular targets once certain clickable functional groups are in proximity to each other[27,28]. Clearly, given aforementioned well-established click reactions, and other transformations with advantages and inherent limitations[9,17,29–31], versatile click reactions are still highly in demand, to expand the toolkit and landscape of click chemistry[6,9,30,32].

Light activation is an ideal way to drive chemical transformations over chemical and biological processes, conferring several unique advantages, including operational simplicity without use of toxic metal catalysts and ligands, higher resolution in space and time[32,33]. Recently, light-induced click chemistry with spatial and temporal precision provides fascinating opportunities for both life science and materials science[32–37]. For example, light-induced thiol-alkene/alkyne (thiol-ene/yne) reactions, and tetrazole photoclick reaction have been successfully implemented in materials science and biological systems[32,33]. Although many exciting applications have been found, currently, only a limited number of photoclick methodologies have been developed, and photoclick chemistry is still in its infancy[32–37]. Several challenges are associated with the discovery of light-induced click reactions[32,33]. Most notably, merging photon utilization with chemical transformations to initiate the proposed reactions, selectively and efficiently generating new molecules with desired functions, remains largely elusive in a variety of matrices, especially in complex biological environments. In addition, it is often challenging to design photo-responsible substrates with structure diversity as modular units to fulfill the criteria of click chemistry[32]. Furthermore, the requirement of specialized photoclick handles involving laborious multistep syntheses can often limit reaction generality. Therefore, with growing demands of click reactions combining the intrinsic advantages of photoclick process, developing novel photoclick reactions remains highly desirable and challenging[32,33].

Herein, we have developed the light-induced primary amines and o-nitrobenzyl alcohols (o-NBAs) cyclization (hereafter, PANAC) as a photoclick reaction under operationally simple and mild conditions, enabling rapid functionalization of diverse small molecules, and native biomolecules in vitro and in living systems (Fig. 1b). With intrinsic advantages of temporal control, reliable chemoselectivity, excellent efficiency, and readily accessible reactants, PANAC photoclick chemistry provides versatile platform for modular conjugation of multiple FM and primary amines, one of the most abundant functional groups as straightforward click handle, expanding the click chemistry toolkit.

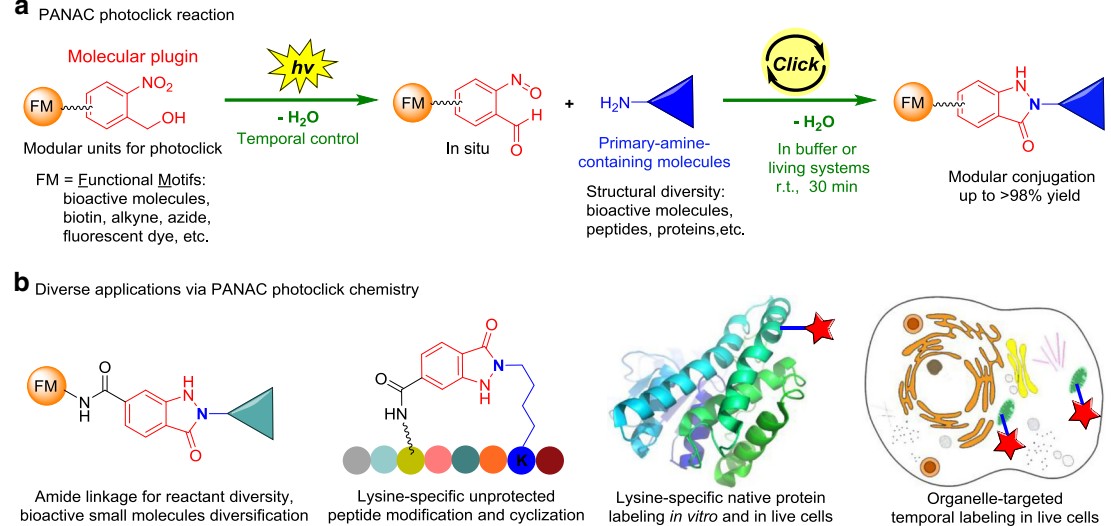

**Fig. 1 Design of light-induced PANAC conjugation enables modular functionalization of small molecules and native biomolecules in temporal control. a** o-Nitrobenzyl alcohol (o-NBA) was designed as molecular plugin and amide formations, as general linkage to rapidly access diverse reactants as modular units, and primary amines as straightforward click handle for PANAC photoclick reaction in vitro and in living systems. **b** Diverse applications of this PANAC photoclick chemistry.

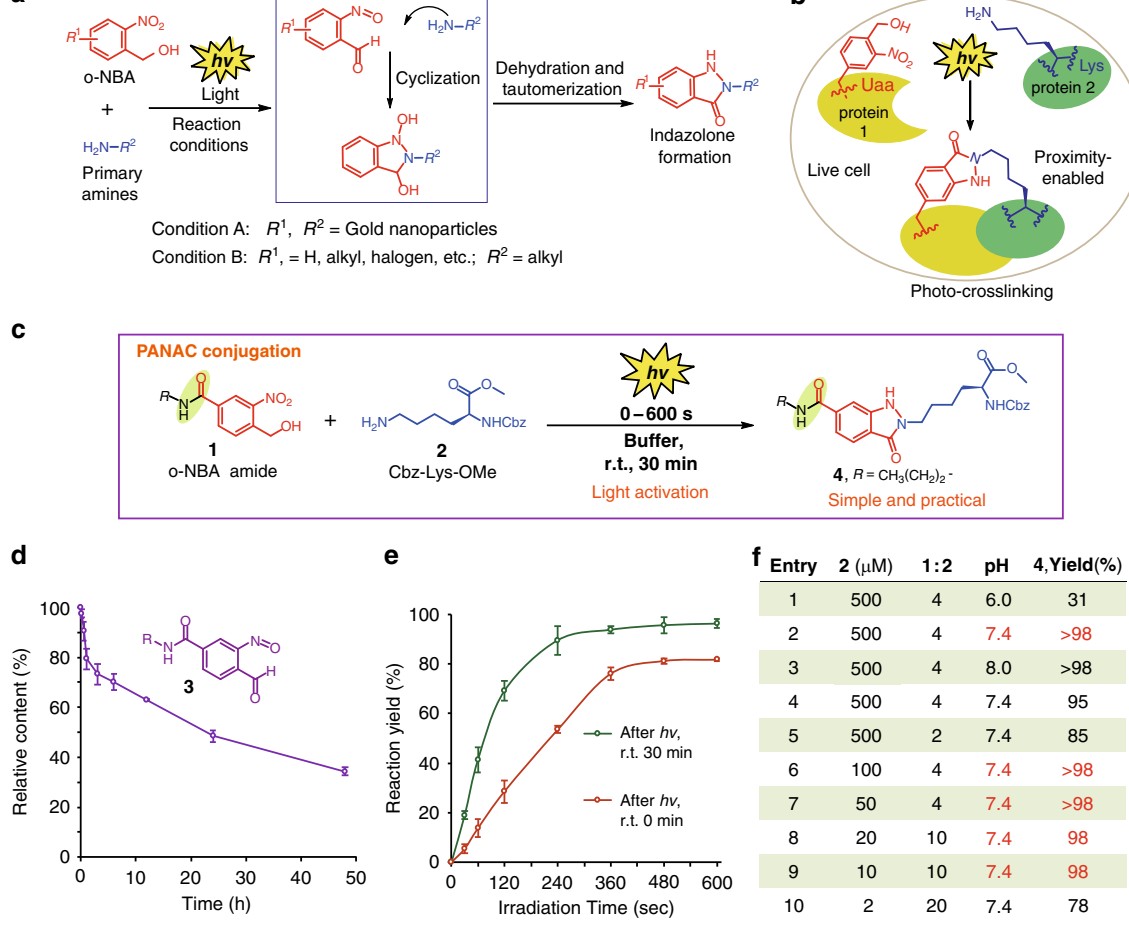

**Fig. 2 Identification of *o*-NBA amide and optimizing conditions for light-induced PANAC conjugation. a** General scheme for previous reported light-induced indazolone formation. **b** The residue-selective photo-crosslinking approach to capture protein–protein interactions in living cells based on proximity-enabled indazolone formation. **c** PANAC conjugation of primary amine **2** with *o*-NBA amide **1**. **d** The stability of the intermediate aryl-nitroso **3** photogenerated from **1** in neutral buffer (100 mM PBS/MeOH = 2:1, pH = 7.4), data are shown as mean ± SEM (*n* = 3 independent experiments). **e** The reactions proceeded smoothly after hundreds seconds of light activation (red line), a further incubation of the reaction mixtures to give increasing yields (green line), data are shown as mean ± SEM (*n* = 3 independent experiments). **f** Optimization of reaction conditions. Yields were determined by ratio of peak area value of experiment to that of internal standard product on reverse-phase HPLC, are reported as an average of three independent trials, see Supplementary Information. With scale of 0.1 mmol (**2**, 2 mM, **1:2** = 4) in **f** of entry 2, the reaction provided **4** with 90% isolated yield.

## Results

**Design of light-induced PANAC conjugation in click chemistry context.** Recently, Zhao and coworkers described the light-triggered indazolone formation from *o*-NBA derivatives and benzylamine for assembly of gold nanoparticles in aqueous solution (Fig. 2a, condition A)[38]. However, the reaction efficiency vs condition, the nature of the *o*-NBA structures and the feasibility of this transformation under complex environments, such as, in complex biological environments for bioconjugations, have not been explored yet remain elusive (Fig. 2a, condition A). Very recently, Kurth[39] and our group[40] independently reported the photochemical synthesis of indazolone heterocycles from *o*-NBA analogs and primary amines via (addition/cyclization/dehydration/tautomerization) mechanistic pathway, with moderate to good yields in different reaction conditions (Fig. 2a, condition B). Whereas, this transformations still suffer from harsh reaction conditions (e.g., long time UV exposure, 3–24 h), and the reaction efficiency (e.g., 3–24 h reaction time) and substrate diversity still need to be improved with respect to synthetic applications. Meanwhile, we developed a residue-selective photo-crosslinking approach to capture protein–protein interactions (Fig. 2b) via indazolone formation from *o*-NBA-derived unnatural amino acid

(Uaa, *o*-NBAK) and ε-amino group of proximal lysine in living cells upon light activation[41]. However, this indazolone formation was achieved by the increased effective concentration via proximity effects based on protein–protein interaction in biological environments (proximity-enabled, Fig. 2b). Although these studies revealed the successful indazolone formation under light activation, the generality (e.g., efficiency, accessibility, and functionality) of abovementioned indazolone formations do not meet the criteria of click chemistry[1,30,32].

In particular, we recognized that the photo-crosslinking transformation is biocompatible, and the *o*-NBA backbones are principally inert before light activation in protein complexes and living systems (Fig. 2b); thus, the *o*-NBA derivatives could serve as general masked reactants and photo-responsible handles in complex environments[41]. Inspired by the photochemical indazolone formations[38–40], and the residue-selective photo-crosslinking technology in living cells[41] (Fig. 2a, b), as well as the abundance of primary amines[42,43], we questioned whether the light-induced PANAC reaction might be successfully evolved as a photoclick reaction (Fig. 1a), in which primary amines—one of the most abundant functional groups—would be ideal and direct click handles, thus for rapid conjugation of diverse

 

primary-amine-containing molecules and functional *o*-NBA motifs, in temporal control and biocompatibility. Consequently, with intrinsic advantages of light-induced spatiotemporal control and reliable chemoselectivity[32,33,41], the light-induced PANAC conjugation might offer many opportunities and potential applications to myriad chemical and biological processes (Fig. 1a)[32,33,43].

To achieve the accessibility of this proposed light-induced PANAC reaction as a click reaction[30], we rationalized that a key element to realize this strategy would be the straightforward protocol to rapidly access diverse functional *o*-NBA reactants, to meet the standard of click chemistry. We envisioned that the *o*-NBA structure could serve as a molecular plugin (Fig. 1a), but a general linkage is necessary for the assembly of this molecular plugin and different FM into corresponding modular units (*o*-NBA reactants). Thereby, we decided to systematically investigate the electronic feature and the position of substituents on the *o*-NBA backbones related to reactivity. After screening of the *o*-NBA analogs, we found that the substituents on the aryl ring of *o*-NBA have a significant effect on the reaction efficiency (Supplementary Table 1). Impressively, *o*-NBA backbone with electron-withdrawing amide group (Fig. 2c, *o*-NBA amide, **1**) turned out to be highly reactive with primary amine **2**, smoothly affording excellent yields. With insight into this structure–reactivity relationship, we considered that the amide bond formation would be an ideal linkage for direct assembly of modular units from *o*-NBA handle (molecular plugin) and multiple FM (Figs. 1b and 2c), to rapidly access a wide range of *o*-NBA derivatives.

**Establishing the optimal procedure for light-induced PANAC conjugation.** To investigate whether the efficiency of PANAC reaction fulfills the criteria of the click reaction[1,32], we examined the stability of the photogenerated aryl-nitroso intermediate in PANAC reaction and optimized the reaction conditions. Notably, with a quantum yield of 0.52 (Supplementary Figs. 16 and 17)[38], the photogenerated intermediate aryl-nitroso **3** proved relatively stable in neutral buffer ($t_{1/2} \geq 20$ h) rather than minutes (Fig. 2d), which highlighted the potential for general reactivity without compromising the reaction efficiency, whereas certain photogenerated intermediates can be readily quenched in photoclick process[32,33]. We conducted a survey of the reaction efficiency with different time range of light activation and the resultant incubation of reaction mixture. The reaction is rapid to provide high yields after hundreds seconds of light activation (Fig. 2e), and a further incubation (r.t. 30 min) of the reaction mixture is better to give increasing yields. Studies on the reaction kinetics (Supplementary Fig. 18) revealed that the second-order rate constant reaches 87.4 M/s, which is fast and comparable to those of CuAAC, tetrazole photoclick, and certain tetrazines involved reactions[29]. Further conditions screening found that pH values of buffer are very critical for the reaction efficiency. Buffers with pH 7.4 or above (Fig. 2e, entries 2 and 3) provided almost quantitative yields (up to >98%) in 30 min, while buffer with pH 6.0 resulted in low yield (Fig. 2e, entry 1). Importantly, the light-induced PANAC conjugation tolerated a range of reaction conditions (e.g., different buffer, additives, Supplementary Table 2). Alternatively, the reaction also proceeded rapidly to afford excellent yield in other conditions (Fig. 2e, entry 4), where the *o*-NBA amide **1** was activated with light at 365 nm for 7 min before adding to the sample mixtures. Thereby, this reaction conditions are suitable and potentially highly valuable for light-sensitive biological samples[23]. Notably, reducing the concentrations of **2** to 2 μM (Fig. 2e, entries 6–10) still provided good yields in 30 min, which indicated the reaction is highly efficient for conjugation of low abundance biomolecules, such as native proteins in complex

biological environments[9,44,45]. It is also worth mentioned that *o*-NBA was used in low amount (Fig. 2e, entry 5, 2 equiv.), the reaction still proceeded smoothly with high yield.

We next examined the chemoselectivity of this light-induced PANAC conjugation. Indeed, when an equimolar mixture of **2** and amino acid with nucleophilic sidechain were treated with **1** upon light activation, no detectable ligation products were found from the reactions of **1** and different nucleophilic sidechains in competition experiments (Supplementary Table 3, Fig. 1 and Supplementary Figs. 28–34). Finally, the PANAC product was hydrolytically and thermally stable in buffer conditions (Supplementary Fig. 2).

Taken together, these results indicated that the *o*-NBA amide reacted sufficiently fast and highly chemoselective with primary amine upon only several minutes light activation, providing near-quantitative yields under the optimal reaction conditions. Compared to the previous reported harsh conditions for indazolone synthesis under long time UV exposure[38–40], and photo-crosslinking based on proximity-enabled reactivity[41] (Fig. 2a, b), we demonstrate that the electron-withdrawing amide group of *o*-NBA reactant (Fig. 2c), and the buffer conditions (Fig. 2f) are critical for the reactivity and efficiency in PANAC reaction. Especially, the light-induced PANAC reaction is highly efficient with low concentrations of reactant (2–500 μM) under operationally simple and mild conditions, which is not sensitive to oxygen or water without the need of catalyst, thus revealing the feasibility to perform PANAC conjugation in complex environments and click chemistry context.

**Light-induced PANAC reaction for modular conjugation of molecules with multiple functional groups.** To further evaluate whether the chemoselectivity and generality of the light-induced PANAC reaction meets the standard of click chemistry, we investigated the reaction of various primary-amine-containing molecules with different types of click partners for assembly of molecules containing multiple functional groups[10,46]. As highlighted in Fig. 3, various functional groups were tolerated on the both of reaction partners. To our delight, *o*-NBA amides showed high reactivity toward different primary amines, smoothly affording products with excellent yields (Fig. 3, **5**–**12**, up to >98% yield). We examined PANAC conjugation for direct late-stage diversification of pharmaceuticals using various known molecules (mafenide, tyramine, lenalidomide analog, E7820 analog, linezolid, amlodipine, doxorubicin, and 3′-amino-3′-deoxythymidine). Surprisingly, in mild conditions and on short timescales (30 min), we could achieve desired products with high efficiency (Fig. 3, **13**–**20**). Furthermore, the PANAC conjugation of different *o*-NBA amides containing functional groups (biotin, fluorescent dye, and azide) provided products in near-quantitative yields (Fig. 3, **21**–**23**). Finally, we applied this PANAC conjugation strategy for rapid assembly of proteolysis targeting chimeras[47] (PROTACs, Fig. 3, **24**–**25**), which are bispecific molecules containing a ligand of E3 ubiquitin ligase and a target protein binder connected by *o*-NBA amide. This strategy allows a rapid and parallel synthesis of libraries of PROTACs for inducing protein degradation, an emerging therapeutic strategy for undruggable targets with exciting prospects.

In the development of photoclick reactions, it is often challenging to design diverse photo-responsible substrates as modular units, to fulfill the criteria of wide scope reactants in click chemistry[32]. Indeed, the amide formation turned out to be a practical and general linkage to rapidly access *o*-NBA derivatives, as photo-responsible modular units in PANAC conjugation (Fig. 3). These results revealed the successful execution of our design ideas to assembly modular NBA

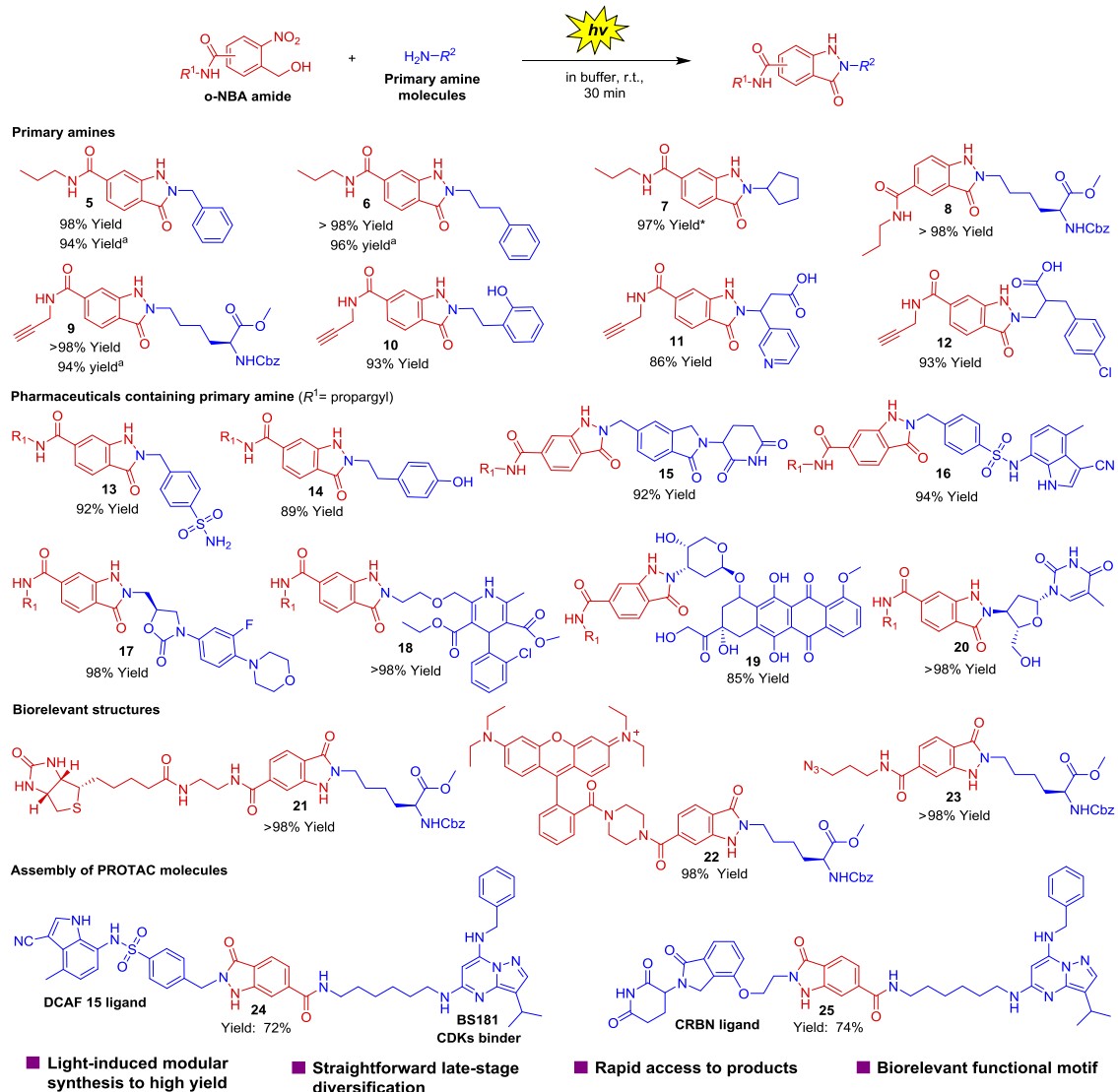

**Fig. 3 Light-induced modular functionalization of primary amines, straightforward late-stage diversification of pharmaceuticals, and rapidly assembly of PROTACs.** Reaction conditions: amine (500 μM) and $o$-NBA derivatives (2 mM) in 100 mM PBS/MeOH (2:1, pH = 7.4) were treated with 365 nm UV light for 7 min and incubated at 25 °C for 30 min. *The concentration of $o$-NBA amide was 4 mM. For compound **10/11/13**, the substrates were irradiated for 10 min. For compound **22**, the substrates were irradiated for 15 min. Yields were determined by ratio of peak area value of experiment to that of internal standard product on reverse-phase HPLC, are reported as an average of three independent trials, see Supplementary Information for HPLC trace. [a]With the scale of 0.1 mmol amine, isolated yields.

reactants from molecular plugin and FM via a general linkage (amide formation, Fig. 1d). More importantly, with the unique mechanistic pathway[40] (Fig. 2a), the light-induced PANAC conjugation presents excellent selectivity to primary amines groups, while being orthogonal to other common functional groups, including alcohols, phenols, carboxylic acids, secondary and tertiary amines, and nitrogen heterocycles in synthetic chemistry and conjugations (Fig. 3, **10–25**). Taken together, these results indicate that PANAC conjugation enabled modular functionalization of small molecules via primary amine group as direct click handle, providing rapid access to diverse molecules without laborious de novo chemical synthesis, which would find broad applications in organic synthesis[10], medicinal chemistry[46,47], and chemical biology[28].

**Light-induced PANAC conjugation for labeling and cyclization of unprotected peptides with temporal control**. We next verified the ability of light-induced PANAC conjugation in peptide

chemistry[44,48], to evaluate its diverse chemical function as a click reaction[1]. We evaluated this light-induced conjugation approach using small peptides to assess the applications in peptide labeling. We prepared the $o$-NBA backbone containing tags (biotin, azide, and alkyne), and peptides with lysine and other amino acids with potentially different nucleophilic side-chains (Fig. 4a). The light-induced peptide conjugations turned out to be very rapid, to give desired products with up to >98% conversion determined by UPLC–mass spectrometry (MS) analysis of the reaction mixture, respectively (Fig. 4a, **26–28**). This peptide labeling proved to be lysine specific; however, several frequently used lysine-reactive electrophiles revealed promiscuous reactivity[27,28]. These results demonstrate that the light-induced lysine-specific peptide labeling with excellent efficiency and operational simplicity, offering a straightforward avenue of great importance to functional peptide conjugates, whereas incorporation of click handles is required for peptide conjugates when employing some click reactions[49].

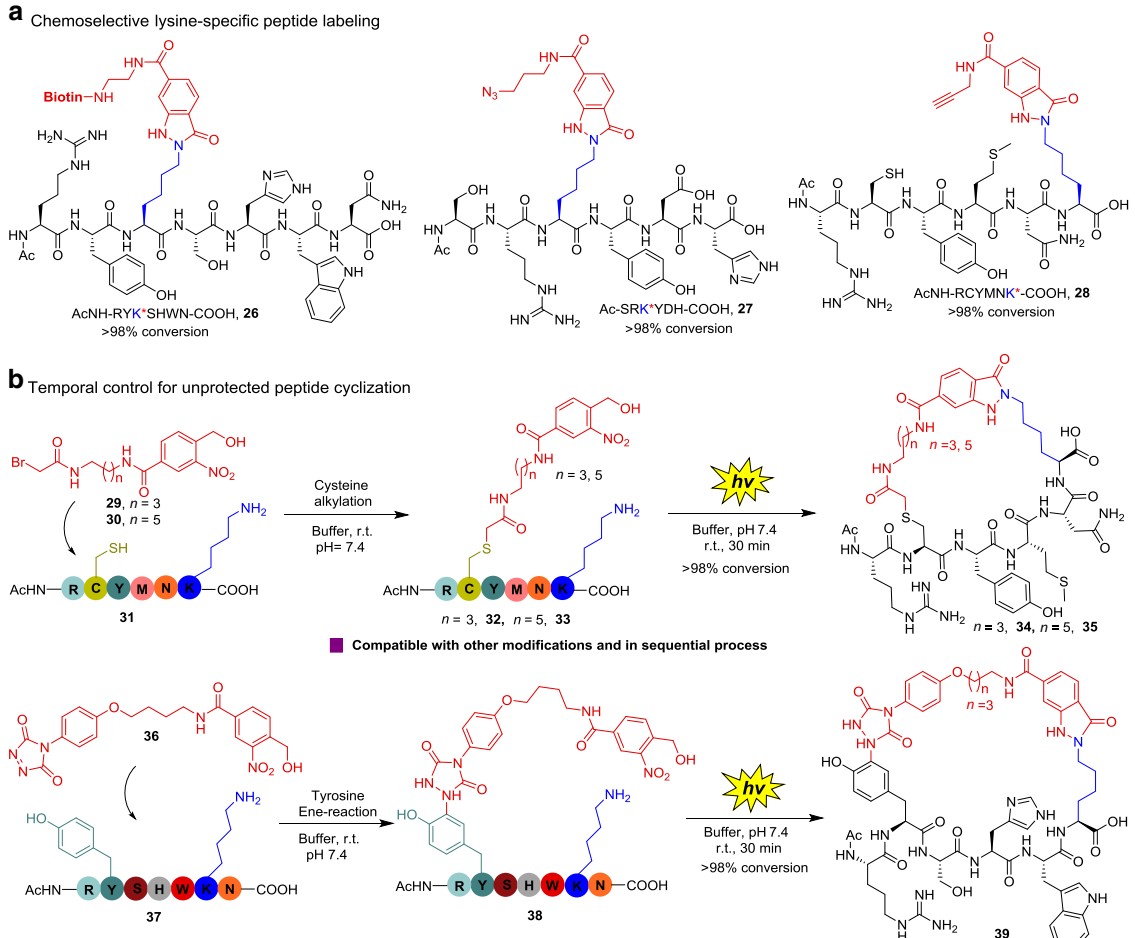

**Fig. 4 Light-induced chemoselective and temporal control of unprotected peptide labeling and cyclization. a** Lysine-specific unprotected peptide labeling, reaction conditions: peptides and *o*-NBA derivatives in 100 mM PBS/MeOH (9:1, pH = 7.4) were treated with 365 nm UV light for 15 min and incubated at 25 °C for 30 min, conversions were determined by reverse-phase HPLC as an average of three independent trials, see Supplementary Information for HPLC trace. For cyclization peptide **28** DTT was used for reducing cysteine dimerization after the reaction. **b** Lysine-specific unprotected peptide cyclization, see Supplementary Information for HPLC trace, conversions were determined as in **a**.

To further explore the potential ability of the temporal control in light-induced PANAC conjugations, we applied this temporally controlled conjugation approach for unprotected peptide cyclization[50]. The bifunctional reagents **29–30** and **36** (Fig. 4b) were prepared. After the cysteine residue addition with **29** or **30** in buffer[44], the reaction mixture was directly treated with light and the cyclization of peptide **32** to desired peptide **34** proceeded rapidly, affording up to >98% conversion determined by UPLC–MS analysis, as well as that of peptide **33** to cyclic peptide **35** (Fig. 4b). Similarly, the light-induced direct lysine cyclization of peptide **38** produced from tyrosine ene-type reaction of peptide **36** with **37** (refs. [22,44]), to cyclic peptide **39** smoothly proceeded again, with up to >98% conversion (Fig. 4b). These results showed that light-induced temporal control of lysine-specific cyclization on unprotected peptides is compatible with other residue-selective chemical modifications and the multiple modifications can be performed in sequential process, without the protecting and deprotecting procedures for certain sidechains of amino acids, providing a unique and direct protocol for synthesis of cyclic peptides as potential medicinal agents[50]. By contrast, highly lysine-reactive electrophiles (e.g., activated ester and aryl sulfonyl fluoride) revealed promiscuous modifications with other nucleophiles (e.g., O, N-nucleophiles) under complex environments[27,28]. Cyclization of peptides with residue selectivity has emerged as attractive strategies for developing therapeutic

agents and biochemical tools[50]. Therefore, the PANAC conjugation clearly demonstrate the operational simplicity, reliable selectivity, and excellent efficiency for lysine-specific labeling and cyclization of unprotected peptides, providing an efficiently chemical route to construction of the cyclic peptide motifs[50], thus, expanding the synthetic toolbox for peptide stapling[51], proteomimetics, and medicinal chemistry[46].

**Light-induced PANAC conjugation for labeling native proteins in vitro.** To validate the functionality of the light-induced PANAC reaction as a click reaction for the bioconjugation of multiple functionalities on native biomacromolecules in vitro, such as native proteins rather than small peptides, human epidermal growth factor receptor type 2 (HER2)[52]-specific nanobody (nanobody–HER2) was chosen as model protein (Fig. 5a). The light-induced lysine-specific labeling of nanobody–HER2 was completed in 30 min. Covalent modification of nanobody–HER2 was confirmed by ESI-TOF MS analysis and gel-based assay (Fig. 5a–c). Indeed, ESI-TOF MS analysis of nanobody–HER2 conjugate indicated almost quantitative modification (Fig. 5b and Supplementary Fig. 3a). Trypsin digest and subsequent ESI-MS/ MS analysis of the fragments confirmed the lysine-specific labeling (Fig. 5d, and other fragments see Supplementary Fig. 3b). The SDS–PAGE fluorescent gel (Fig. 5a, c) also

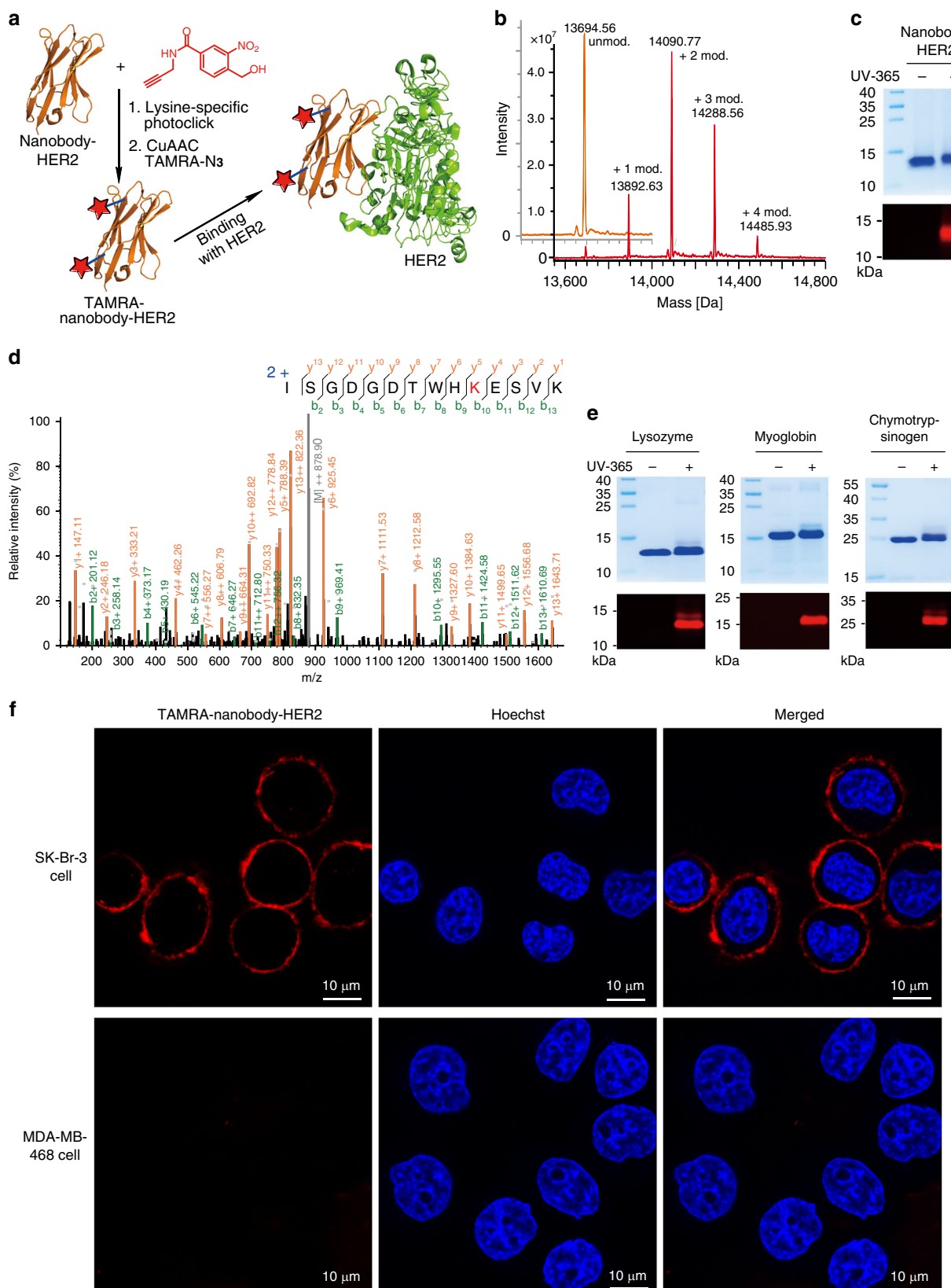

demonstrated light-induced PANAC conjugation of *o*-NBA-alkyne (Supplementary compound **S14**) with lysine is light-dependent and compatible with subsequent CuAAC. It is worth noting that this light-induced lysine-specific PANAC conjugation is also robust for other native protein modifications, such as

myoglobin, lysozyme, and α- chymotrypsinogen A in identical conditions (Fig. 5e and Supplementary Figs. 4–7). Altogether, these results indicate that the light-induced PANAC conjugation is lysine specific, and highly efficient for native proteins modification and functionalization. Many antibody–drug conjugates

**Fig. 5 Light-induced temporal labeling of native proteins with lysine-specific and nanobody–HER2 labeled with TAMRA for live cell imaging. a** General scheme for nanobody–HER2 labeling and binding with HER2 extracellular domain (green; PDB ID: 5MY6). **b** ESI-TOF MS analysis of unmodified nanobody–HER2 and light-induced labeling of nanobody–HER2 with *o*-NBA-alkyne, and unmodified nanobody–HER2 is ~2%. Unmod. refers to unmodified; mod. refers to modified. **c** Nanobody–HER2 labeling with *o*-NBA-alkyne from **b** was subjected to CuAAC with TAMRA-N$_3$, in CuAAC, partial (~20%) modified nanobody–HER2 protein aggregation and precipitation was observed, then analysis of SDS–PAGE gel stained with Coomassie blue (top) and fluorescence image (bottom). **d** Tandem MS analysis of nanobody–HER2 labeling with *o*-NBA-alkyne. **e** SDS–PAGE gel analysis of light-induced labeling of myoglobin, lysozyme, and α-chymotrypsinogen A, for tandem MS analysis see Supplementary Information. **f** SK-Br-3 cells were stained with TAMRA-labeled nanobody–HER2 (red) and Hoechst 33342 (blue) at 37 °C for 1 h and 15 min, respectively, followed by PBS washing three times and then imaging using confocal fluorescence (top), scale bar 10 μM. MDA-MB-468 imaging using identical conditions (bottom).

(ADCs) under clinical evaluation are developed using lysine-based conjugation strategies (e.g., FDA-approved Mylotarg and Kadcyla)[53], despite the potential of heterogeneous products. Our PANAC conjugation demonstrated near-quantitative modifications for several native proteins (up to 98% modified, Fig. 5b, Supplementary Figs. 3a and 4), while prevalent lysine-based conjugations (e.g., activated esters and imido ester) still provide unconjugated antibodies, such as ~50% unconjugated antibody in Mylotarg[53]. Clearly, our currently developed PANAC conjugation offers a rapidly chemical strategy to the development of protein drugs and therapeutic antibodies, with multiple functionalities[54].

Next, we examined the binding ability of the labeled nanobody with target, to assess the applicability of PANAC labeling strategy in protein modification. We used the fluorescent-labeled TAMRA–nanobody–HER2 (Fig. 5a–c) to examine HER2 imaging in different breast cancer cell lines. Strong fluorescence from the HER2-positive breast cancer cell line (SK-Br-3) imaging indicated that the high expression of HER2 and good affinity of labeled nanobody binding with HER2 antigen. In contrast, no fluorescence was observed from the HER2-negative breast cancer cell line (MDA-MB-468) in identical protocol, due to the low expression of HER2 on this breast cancer cell membrane[44]. These results indicated that modified nanobody–HER2 binds original HER2 target well with sensitivity and specificity via PANAC conjugation, while introduction of the fluorescence dye allowed the nanobody conjugates for breast cancer imaging or diagnostics. The sensitivity and specificity of modified proteins to the targets or antigens are critical for the assessment of the labeling strategies, which have emerged as very important tools for engineering novel biological diagnostics, imaging probes, and therapeutics in biomedical research[44]. Taken together, with residue specificity, high efficiency, and reliable binding ability, our currently developed PANAC conjugation suggests a direct avenue for diagnostic nanobody with multiple functionalities, leading to design and development of novel ADCs, biological diagnostics, imaging probes, and therapeutics[44,45,54].

**Light-induced PANAC bioconjugations for native biomolecules in living systems**. Finally, we evaluate the biocompatibility of the PANAC approach and executed it for bioconjugations in living systems[33]. We wondered if light irradiation conditions might cause sufficient harm to cells. Fortunately, the cell availability assay showed that the short time of 365 nm UV irradiation employed in PANAC reactions, as well as in the presence of (1 or 4 μM) *o*-NBA-derived probes, did not noticeably affect cell proliferation of MDA-MB-468 cells (Supplementary Fig. 21). We applied the light-induce PANAC strategy for kinase profiling with lysine-reactive chemoproteomic probe in live mammalian cells (Fig. 6a–c). Structural analysis indicated that each of the 500+ human protein kinases has a catalytic lysine in the ATP binding site[28]. The pyrimidine 3-aminopyrazole scaffold (Fig. 6a, b) which can form three hydrogen bonds with kinase conserved hinge region and has been shown binding with many kinases[55]. Based on the typical crystal structure of protein kinase domain (e.g.,

STK24, PDB: 4QO9) in complex with inhibitor, the docking modes (Fig. 6b) suggested that the kinase domain is capable of accommodating probe-1 and photogenerated nitroso-probe-1, implying that the *o*-NBA structure of probe-1 could react with the conserved lysine at the ATP binding site upon light activation. We then synthesized the probe-1 and showed that this probe can capture up to 91 kinases from Jurkat cell line and 76 kinases from K562 cell line, of which 39 kinases were identified in both cell lines (Fig. 6c and Supplementary Table 4). The recently reported photoreactive diazirine pan-kinase probes without residue specificity only can detect up to 22 intracellular kinases[56]. These results indicated that lysine-targeted chemoproteomic probe is biocompatible, cell-based and temporally controlled, enabling broad-spectrum kinase profiling in live cells, thus could be used in assessment of clinically relevant kinase-inhibitor occupancy[28]. In fact, the two-step diazotransfer/CuAAC click chemistry for modification of primary amines in living systems remains to be explored[10]. Altogether, our strategy offers a complementary approach for addressing the challenge of kinase–drug interactions in drug discovery, otherwise not accessible with kinobeads or ATP–biotin probes in live cells[55].

Based on the success of PANAC bioconjugation for native protein labeling in living cells, we sought to explore whether our conjugation strategy might be implemented to allow the temporally controlled organelle-targeted labeling beyond the scale of native proteins. Thus, we applied this strategy for mitochondria-selective labeling and imaging in living mammalian cells (Fig. 6d, e). The mitochondria-targeted fluorescent probe (Rho-*o*-NBA) consists of *o*-NBA moiety and rhodamine B group, which is widely used to target mitochondria[57] (Fig. 6d). The uptake of Rho-*o*-NBA probe and the representative mitochondrial probe Rhodamine 123 (Rho 123) into cells, and analysis of subcellular localization were performed using confocal imaging of the breast cancer cells (MDA-MB-468). After cells were co-incubated with Rho-*o*-NBA probe and Rho 123, confocal imaging analysis showed that Rho-*o*-NBA mitochondria localization merged well with that of Rho 123 with or without light activation (Supplementary Fig. 20a and Fig. 6e top panel, with high Pearson's correlation coefficient = 0.93), which indicated that developed probe Rho-*o*-NBA efficiently target mitochondria in living cells. In addition, cells transfected with an EGFP reporter plasmid with specific mitochondria localization (Mito-EGFP)[58], then incubated with Rho-*o*-NBA probe followed with light activation, indicating the good enrichment and confine of Rho-*o*-NBA probe in the mitochondria organelles in live cells (Fig. 6e, bottom panel). To confirm that the Rho-*o*-NBA probe could form a covalent bond with mitochondria after photoclick, we used the non-covalent probe Rho 123, then fixed, and washed the cells using washing buffer to remove unreacted Rho 123 for intended comparison (Supplementary Fig. 8). Indeed, Rho-*o*-NBA probe also showed strong mitochondrial localization, while the Rho 123 could not provide signals due to the non-covalent Rho 123 completely washed away. Moreover, these reaction-based covalent labeling of the mitochondria is light-dependent event (Supplementary Fig. 8). Of note, due to good cell uptake and subcellular

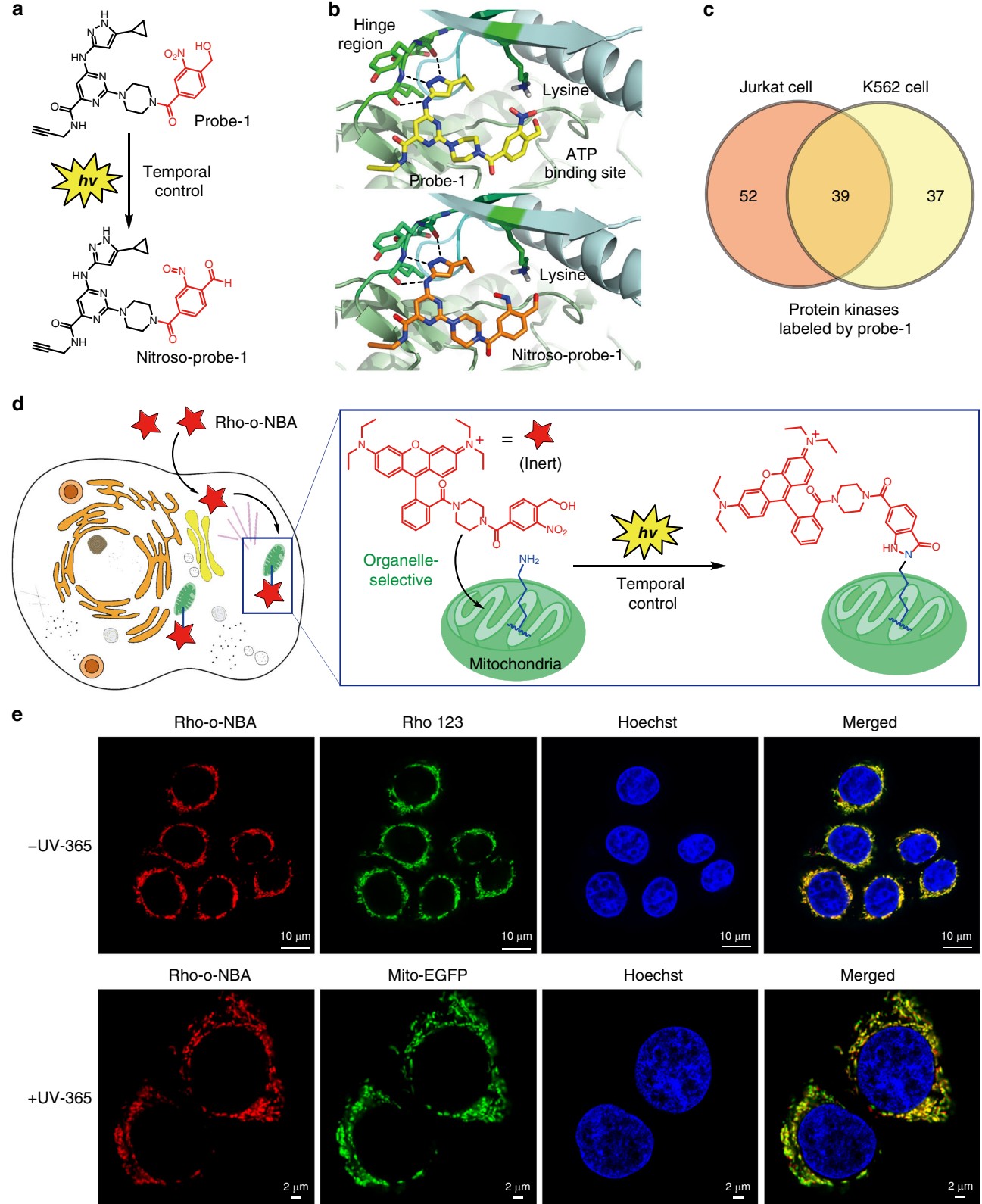

distribution (Fig. 6e, top panel) when using Rho-*o*-NBA probe, it is not necessary to fix cells for regular imaging or dynamic mitochondrial tracking in living mammalian cells, but light activation procedure can be added to achieve further temporal covalent labeling. On the other hand, as spatial control for live cell labeling (Supplementary Figs. 9–12, 13–15), we have demonstrated the cell surface labeling with the HER2-specific nanobody (Fig. 7a)

and mitochondria-targeted labeling (Fig. 7b) via PANAC photo-click reaction, simply by shielding part of the cells from light irradiation (bottom panel), or with UV-365 nm light activation (top panel). This unique covalent labeling ability is highly valuable and should be readily expanded to other organelles[45] with targeted motifs in a temporally and spatially controlled manner, such as tracking organelle dynamics from upstream signaling[28,59] in

**Fig. 6 Light-induced profiling of endogenous kinases and mitochondria-targeted labeling in live cells. a** Design of probe-1 targeting ATP binding site of kinase domain. Nitroso-probe-1 was photogenerated with temporal control. **b** Docking mode of STK24 kinase domain (PDB code:4QO9) with probe-1 (top), with photogenerated nitroso-probe-1 (bottom). **c** Venn diagrams showing the number of shared and unique kinases labeling with probe-1 from Jurkat cells and K562 cells, identified by LC–MS/MS after treatment of cells with probe-1. Captured protein kinases identified with ≥1 unique peptide, see Supplementary Information for details. **d** Schematic illustration of mitochondria-targeted temporal labeling via light-induced bioconjugation with fluorescent Rho-*o*-NBA probe in living cells. **e** Confocal images of MDA-MB-468 cells stained with Rho-*o*-NBA probe (red channel), Rhodamine 123 (green channel), and Hoechst 33342 (blue channel) in different conditions: colocalization of Rho-*o*-NBA probe and Rhodamine 123 without light activation (top), colocalization of Rho-*o*-NBA probe, and Mito-EGFP reporter under light activation (bottom), without light activation see Supplementary Fig. 20b.

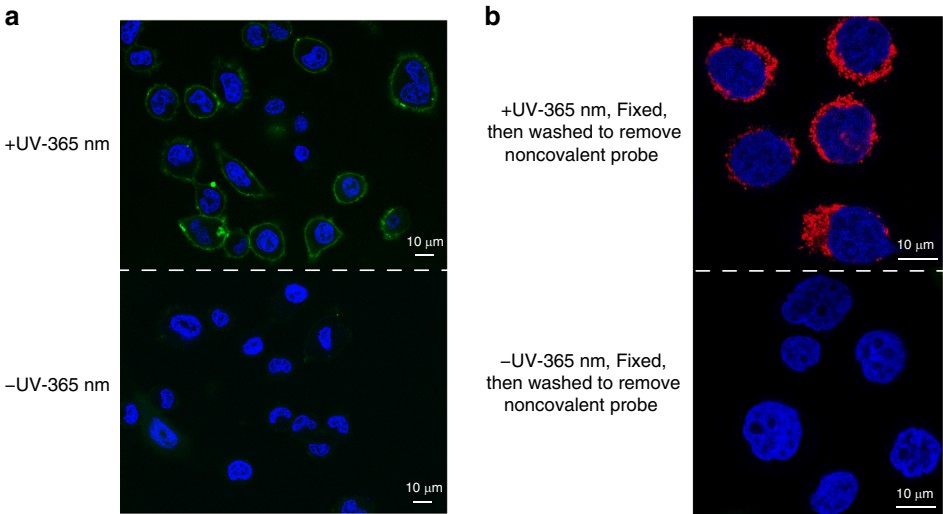

**Fig. 7 Spatial labeling of live cells via PANAC photoclick reactions. a** Labeling of HER2-specific nanobody sdAb-HLC-*o*-NBA, with FITC-NH$_2$ on the surface of SK-Br-3 cells, simply by shielding part of the cells from light irradiation (bottom panel), or with UV-365 nm light activation for 10 min (top panel). After labeling, cells were washed by PBS wash for three times, and cell imaging using confocal fluorescence, see Supplementary Fig. 9 for details.
**b** Confocal images of MDA-MB-468 cells stained with Rho-*o*-NBA probe (red channel) and Hoechst 33342 (blue channel) in different conditions: Rho-*o*-NBA probe with light irradiation (top panel), or shielding part of the cells from UV-365 nm light irradiation (bottom panel), then fixed with 4% PFA, and washed to remove unreacted probe for comparison. Scale bar: 10 μm.

response to exogenous ligand-induced activation or stimulus-dependent event in living cells, otherwise not accessible with traditional spontaneous thiol-reactive probes (e.g., organelle-localizable reactive probes, Mito Tracker probes)[45,57]. Collectively, we demonstrated that the PANAC conjugation is well biocompatible and applicable to label native proteins further subcellular organelle, opening up an avenue for probing and functionalization of native biomolecules in temporal and spatial control for better understanding of biology in living systems.

## Discussion

By design of *o*-NBA derivatives with amide linkage as modular reactants to achieve the accessibility, establishing the optimal procedure for high efficiency, systematic investigation of reactivity, and diverse chemical function of the reaction, we have developed light-induced PANAC reaction for modular conjugation of diverse molecules via primary amines as direct click handles, in mild buffer conditions and complex environments. We demonstrate that the PANAC conjugation is sufficiently fast and highly efficient for straightforward late-stage diversification of pharmaceuticals and biorelevant molecules, lysine-specific labeling, and cyclization of unprotected peptides as potential medicinal agents, functionalization of diagnostic nanobody as imaging probe, modification of other native proteins in vitro. The generality of PANAC conjugation was further validated by broad-spectrum profiling endogenous kinases and organelle-targeted temporal labeling in living systems.

The currently developed light-induced PANAC conjugation possesses the following unique features, especially in complex

biological environments. (I) Good biocompatibility with fast kinetics. The light-induced PANAC conjugation proceeds within minutes (Fig. 2e) without the use of toxic metal catalysts and ligands in vitro and in living systems, conferring less toxicity to biological systems, compared to certain metal-catalyzed click reactions[17,29,33]. (II) Readily available and modular reactants. Primary amines are one of the most abundant functional groups and readily available[42,43]. The amide linkage proved to be a practical and efficient strategy to assembly of *o*-NBA structure (as molecular plugin) and different FM into wide scope of *o*-NBA reactants as modular units (Fig. 3), thus for modular functionalization of multiple molecules via primary amine group as direct handle. (III) Temporal control. The *o*-NBA derived reactants are inert before light activation (Figs. 4 and 6), thus could implement in time-resolved manner in response to exogenous manipulations, such as ligand-induced activation or stimulus-dependent event in living cells[59]. By contrast, among well-established spontaneous primary amine conjugation approaches[43], highly amine-reactive electrophiles (e.g., activated ester and aryl sulfonyl fluoride) are probably too prone to nonenzymatic or enzymatic hydrolysis with reduced conjugation efficiency, during in vitro conjugations or during the process into cells as chemical biology probes[27,28]. (IV) Exquisite chemoselectivity and high efficiency. The light-induced PANAC conjugation demonstrates excellent orthogonality to other common functional groups (Figs. 3 and 4), and still proceeded smoothly at low concentration (e.g., 10 μM) of primary amine (Fig. 2f). Several frequently used amine (or lysine)-reactive electrophiles cross-react with other nucleophiles (e.g., nucleophilic chains of amino acids, O, N-nucleophiles and

GSH) giving promiscuous reactivity[27,28]. We show here that *o*-NBA-derived reactants are able to overcome several aforementioned challenges in complex biological environments, such as background labeling from spontaneous conjugations or cross-reactivity with other nucleophiles, and reduced efficiency from hydrolysis[27,28]. (V) Direct labeling native biomolecules. There are few examples of photoclick reactions for labeling of endogenous or native biomolecules, only an exception of light-induced thiol click for cysteine residue of native proteins in vitro[44]. Our strategy offers a complementary approach to label native biomolecules in vitro and to probe biological processes in living systems (Figs. 4–6)[9]. Collectively, with these unique features, including operational simplicity and mild conditions in vitro or in living systems[1,7,32], the currently developed light-induced PANAC reaction fulfills the criteria of click chemistry. Therefore, we have demonstrated the light-induced PANAC reaction as a photoclick reaction, to expand the toolkit of click chemistry.

Primary-amine-containing molecules are abundant and widespread in synthetic chemistry, biological systems, and materials science[28,42,43]. With abovementioned features, the light-induced PANAC conjugation would find broad applications in various fields. This strategy could provide particularly attractive and straightforward access to modular synthesis of a large family of bioactive indazolone derivatives (Fig. 3), to direct diversification of natural products via primary amine handles, to rapidly parallel assembly of PROTACs as degraders (Fig. 3) for undruggable targets, thus, dramatically expanding compound libraries and the synthetic toolbox of medicinal chemistry[46]. In addition, lysine is one of the most prevalent residues in the peptides and proteins[28]. It is attractive and practical to apply this approach in the future to develop cyclic peptides[50] and stapling peptides[51], as medicinal agents (Fig. 4), protein-based biological diagnostics, and imaging probes (Fig. 5)[44,45,54]. Moreover, *o*-NBA probe is inert with other cellular nucleophilic species during the process to the targets or organelle before light activation (Fig. 6); therefore, the temporal control of light-induced PANAC conjugation should afford broad utilities, such as development of in situ chemistry for target-guided modular assembly of binding fragments[14], lysine-targeted covalent inhibitors, and global profiling of lysine reactivity in response to upstream signaling in time-resolved operations[28,59], target-selective, or site-specific modifications of native proteins (Fig. 6b)[55,56]. Finally, based on reaction mechanism, the current light-induced PANAC reaction can in principle be extended to other linkages to assemble *o*-NBA handles, with different FM as click reactants (Fig. 1d). Alternatively, *o*-NBA-related handles could be in principle incorporated in native biomolecules, such as proteins (Supplementary Figs. 9–12, 13–15)[18,41], followed by exogenous primary amine probe labeling in spatial and temporal control in vitro or in living systems, which would greatly expand the power of this light-induced PANAC conjugation[9].

In summary, we have successfully developed the light-induced PANAC reaction as a photoclick reaction, providing a versatile platform for direct functionalization of small molecules and labeling of native biomolecules, via primary amines as direct click handles in vitro and in living systems. As the light-induced PANAC conjugation is very simple and practical under mild conditions, we believe this strategy will be easy for researchers to implement in multidiscipline fields. Given the intrinsic advantages of temporal control, good biocompatibility, exquisite chemoselectivity, high efficiency, wide scope in reactants, and operational simplicity in primary amine modular conjugation processes, this PANAC photoclick chemistry would provide a powerful and reliable chemical tool for synthetic chemistry, bioconjugation, medicinal chemistry, chemical biology, as well as materials science.

## Methods

**PANAC conjugation of different *o*-NBAs and primary amines**. *o*-NBA reactants were prepared as a 10 mM stock in MeOH solution. Prime amine reactants were prepared as a 10 mM stock in MeOH or DMSO solution. *o*-NBA reactants (2 mM) and prime amine reactants (0.5 mM) in 100 mM PBS/MeOH (2:1, pH = 7.4) were treated with 365 nm UV light for 7 min, and incubated at 25 °C for 30 min. The samples were collected, diluted with MeOH/H$_2$O, and analyzed by UPLC–MS. The yield reported as an average of three independent trials, which was quantified by ratio of peak area values of reaction products to standard products.

**The cyclization of unprotected peptide Ac-RCYMNK-OH with compound 29**. Compound 29 was prepared as a 2.6 mM stock in MeOH solution. Peptide Ac-RCYMNK-OH was prepared as a 10 mM stock in H$_2$O solution. Compound 29 (0.52 mM) and peptide Ac-RCYMNK-OH (0.5 mM) in 100 mM PBS/MeOH (9:1, pH = 7.4) were incubated at 25 °C for 1 h. The mixture was treated with 365 nm UV light for 15 min and then was incubated at 25 °C for 30 min. The samples were collected, diluted with MeOH/H$_2$O, and analyzed by UPLC–MS. Conversions were determined by reverse-phase UPLC as an average of three independent trials.

**Fluorescence imaging of live cell with modified TAMRA–nanobody–HER2**. *o*-NBA-alkyne (Supplementary compound S14) was prepared as a 2.5 mM stock in MeOH solution. TAMRA-N$_3$ (Sigma) was prepared as a 1.25 mM stock in DMSO solution. CuSO$_4$ (Sigma) and THPTA (Sigma) were prepared as a 50 mM stock in H$_2$O solution and premixed with a volumn ratio of 1:5. Sodium ascorbate (Sigma) was prepared as a 100 mM stock in H$_2$O solution. Hoechst 33342 (Thermo) was prepared as a 1 mg/mL stock solution in H$_2$O. ONBA-alkyne (2.5 mM) in MeOH was treated with 365 nm UV light for 7 min, then was added to nanobody–HER2 (55 μM, 0.77 mg/mL) in PBS one portion to a final concentration of 125 μM and mixed. The mixture was incubated at 25 °C for 1 h. The samples were collected, diluted with H$_2$O, and analyzed by ESI-TOF. The obtained ONBA-alkyne modified nanobody–HER2 (50 μM) was added TAMRA-N$_3$ (100 μM), premixed CuSO$_4$ (100 μM) and THPTA (500 μM), and sodium ascorbate (5 mM). The mixture was rotated at 25 °C for 1 h. Then excess small molecular impurities were removed and nanobody–HER2–TAMRA conjugate (15 μM in PBS) was obtained by PD-10 desalting column (GE). SK-Br-3 and MDA-MB-468 cells were seeded to Chambered Coverglass (Thermo) and allowed to grow to ~70% confluence. SK-Br-3 and MDA-MB-468 cells were incubated with nanobody–HER2–TAMRA conjugate (300 nM) in Fluorobrite DMEM (Thermo) at 37 °C for 1 h and Hoechst 33342 (1 μg/mL) at 37 °C for 15 min, washed with PBS twice and observed under Leica confocal fluorescence microscope (552 and 405 nm).

**Light-induced mitochondria-targeted temporal labeling and imaging in live cells**. MDA-MB-468 cells were seeded to Chambered Coverglass (Thermo) and allowed to grow to ~70% confluence. ONBA-Rhodamine and Rhodamine 123 (Thermo) were prepared as a 1 mM and 400 μM stock in DMSO solution, respectively. Hoechst 33342 (Thermo) was prepared as a 1 mg/mL stock solution in H$_2$O. MDA-MB-468 was incubated with ONBA-Rhodamine (1 μM) and Rhodamine 123 (400 nM) for 30 min in Fluorobrite DMEM (Thermo) at 37 °C for 30 min and Hoechst 33342 (1 μg/mL) at 37 °C for 15 min, washed with PBS twice, exposed to 365 nm or not, and fixed by 4% PFA. The samples were observed under Leica confocal fluorescence microscope (552, 488, and 405 nm).

## Data availability

All the data are available within the article and its Supplementary Information files or from the corresponding author upon reasonable request.

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

## Acknowledgements

This study was supported by the National Science Foundation of China (NSFC Grants 21778062), the National Science & Technology Major Project "Key New Drug Creation and Manufacturing Program" (No. 2018ZX09711002-008), the Science and Technology Commission of Shanghai Municipality, China (No. 18431907100), and the Strategic Priority Research Program of the Chinese Academy of Sciences "Personalized Medicines-Molecular Signature-based Drug Discovery and Development" (No. XDA12050410). We thank Prof. Bing Yang at the MS facility of ZJU for kind help in tandem MS analysis. We also thank Prof. Zhuo Tang at Chengdu Institute of Biology, CAS for helpful discussions on this work.

## Author contributions

A.-D.G. and D.W. performed optimization of reaction conditions, peptides, and proteins labeling, kinases profiling, mitochondria labeling, and cell culture. H.-J.N. synthesized the probes, substrates, and PROTACs, conducted diversification of biorelevant molecules. K.-N.Y., S.-T.L., and L.F. conducted the pharmaceutical diversifications. B.-S.Z. prepared the bifunctional regents for peptides cyclization. C.F. expressed the native proteins for labeling. H.H. and M.T. analyzed the tandem MS of endogenous kinases profiling. C.P. and R.H. analyzed the mitochondria imaging. X.-H.C. conceived and directed the project, wrote the manuscript with inputs from all authors.

## Competing interests

The authors declare no competing interests.
