## [Peer Review File · Nature Communications]

REVIEWER COMMENTS

Reviewer #1 (Remarks to the Author):

Chen et.al. reported a photo-initiated cyclization reaction of primary amine and o-nitrobenzyl alcohols. The reaction is actually the conjugation reaction of primary amines with aryl-nitroso generated from photolysis of o-nitrobenzyl alcohols upon UV irradiation. As mentioned in Figure 2A and 2B, this UV-triggered fast cyclization reaction has been reported (J. Org. Chem. 2018, 83, 15493; RSC Adv. 2019, 9, 13249) and applied to the lysine-selective photo-crosslinking to capture protein-protein interaction. The authors screen different o-NBA substrates, then they found that o-NBA amide proceeded fast photolysis as well as the following cyclization with primary amine, which took around 30~40 minutes in total to accomplish. The reaction showed good selectivity to primary amine in the lysine side chain. The reaction seemed to have good compatibility with terminal alkynes and azide, which are functionalities used in CuAAC. Lysine-specific modification of peptides or proteins was demonstrated using this reaction. The authors also showed preliminary applications of this reaction in endogenous kinase profiling and mitochondria-targeted labeling. In general, the work is of good quality and important to the field of bioconjugation, but the following issues still need to be addressed:

1. The importance of CuAAC and SPAAC as click reaction is largely originated from the bioorthogonality of the reactions. However, this reaction can not be bioorthogonal since lysine is abundant in biological systems. In this sense, Figure 1 was not very proper. I would suggest the authors add some introduction to other fast and biocompatible ligation reactions, such as the CBT chemistry. Besides, comparison of this reaction to known amine-involved bioconjugation method with respect to their efficiency in protein labeling is necessary.
2. The photo-initiation condition may add some advantages to the reaction if spatial control of the reaction can be demonstrated in a well-designed experiment. The importance of temporal control of bioconjugation reactions is not as significant as spatial control, since the other reaction substrate can be introduced to the system with temporal resolution.
3. Original NMR spectra of the products listed in Figure 3 should be provided in the supplementary data.
4. The authors mentioned kinase profiling using the probe containing the o-NBA structure captured much more kinase species than the work using the probe containing the diazirine photo-affinity moiety. It is hard to understand since the carbene species generated from photolysis of diazirine should have much higher chance to form covalent bonds with adjacent groups. The authors may need to comment on this.
5. The mitochondria-targeted labeling was not really a spatial control realized by the photo-controllable reaction. More convincing results to show the advantage of this reaction to spatial labeling may need photo-irradiation confined to subcellular level.

Reviewer #2 (Remarks to the Author):

Pros.:

1. Via systematic optimization, the developed PANAC reaction now has higher yields for various substrates including drugs targets, peptides and protein substrates via in-vitro chemical modification under different UV irradiation conditions. The selectivity of PANAC reaction toward primary appears to be pretty good. After UV irradiation, the o-nitrosobenzaldehyde amide 3 showed negligible reactivity toward alcohols, phenols, carboxylic acids, and nitrogen-containing heterocycles groups in the presence of primary amine. But there is no data to explain the fate of the intermediate, o-nitrosobenzaldehyde amide 3, toward those endogenous nucleophiles without trapping by primary amines, especially in living systems.

2. In this paper, the introduction of amide groups on para-position of o-nitrobenzyl alcohols has increased the reaction efficacy of the well-established PANAC reaction toward solvent exposed primary amine groups, demonstrating a useful photo-cross-linking tool for studying the pharmacological properties of payload molecules via covalent tagging to native primary amines in biological systems. Since the primary amines are key and abundant residues on biomolecules in live cells, it is obvious that this type of PANAC photo-cross-linking reagents hardly show any bioorthogonality.

Cons.:

1. The novelty of this PANAC reaction is not prominent, it is a continuous study derived from previously research work reported by the same group in which the o-nitrobenzyl alcohol is specifically incorporated on the desired residues of target proteins (Ref. 31). The PANAC reaction was early reported in 2011 (Chem. Commun., 2011, 47, 3822–3824) for covalent photo-assembly of gold nanoparticles, and has been utilized for bio-material modifications with decent spatiotemporal resolution (e.g. Chem. Mater. 2019, 31, 4710–4719, ACS Macro Lett. 2016, 5, 19–23, Angew. Chem. Int. Ed. 2012, 51, 9181–9184, Adv. Mater. 2016, 28, 2724–2730). The photon quantum efficiency and the reaction rate constant of the second imine formation step have been reported (Ref. 28), but it is not investigated carefully in this article; The choice of light wavelength for triggering the photo-transformation of 2 were also investigated previously (Ref. 29, J. Org. Chem. 2018, 83, 24, 15493-15498). However, when applying to live cell studies as a bio-conjugation chemistry, the authors ignored the photo-toxicity of using 365 nm UV irradiation and potential cytotoxicity caused by extensively blocking of vital primary amines.

2. The most important characteristic of a photo-click reaction is the control of when and where to induce covalent bond formation between a pair of bioorthogonal groups (azide-alkyne, tetrazole-alkene, azide-cyclopropenone), not native groups, e.g. primary amine. However, even the developed PANAC reactions does not appear to have spatial controllability which probably is due to the long life-time (Fig. 2d) and insufficient reactivity of o-nitrosobenzaldehyde amide 3 toward primary amine, resulting in its diffusion and nonrestrictive labeling in the cellular environment. The unsatisfied bimolecular reaction rate was also reflected as the higher yields of desired indazolone 4 after prolonged incubation up to 30 mins (Fig. 2c, 2e and 2f without indication of reaction time). It is essential to demonstrate the spatial covalent labeling for a photo-cross-linking tool.

3. The developed PANAC reaction still requires irradiation of 365 nm UV for minimal 7 mins to achieve the moderate to high yield after further 30 mins incubation. The photo-physical properties of o-nitrobenzyl alcohol reagents and the PANAC chemistry were not investigated in this article.

4. The authors claimed that the optimized PANAC reaction is fast and comparable to those of CuAAC, tetrazole photo-click reaction and the tetrazine involved reaction (e.g. J. Am. Chem. Soc. 2016, 138, 5978-5983. 660 nm light triggered IEDDA; photo-SPAAC: J. Am. Chem. Soc. 2017, 139, 24, 8090-8093). But there was no data of reaction rates specified in the article to support this argument.

5. The application of PANAC reaction to conjugate peptide and proteins on hydrogel in 3-dimensional space has been previously validated (Chem. Mater. 2019, 31, 4710–4719). In-vitro modification of primary amines in either drug molecules (Fig. 3) or peptides (Fig. 4) or proteins (Fig. 5) does not represent the advantage of the PANAC conjugation reaction because there are many other methods to achieve this goal without mediation through photo-irradiation (e.g. SuFEx, Bioconjugate Chem. 2017, 28, 1422–1433).

6. In live cell imaging experiment, there was a significant expansion of the Rho-o-NBA signals after 365 nm irradiation in the cytosolic region of cells in comparing with the Rho-o-NBA channel without the photo-irradiation which was colocalized well with the Rho123 signals (Fig. 6e, first lane, red color). This phenomenon suggested the non-specific cross-linking reaction of the photo-

transformed Rho-o-Nitrosobenzaldehyde species toward primary amine residues occurred outside the mitochondria organelles. It is important to add an extra channel via immuno-staining of mitochondrial antigens to identify the specificity of the PANAC photo-cross-linking methods. The fixed cell lysate could also be collected for subsequent in-vitro analysis to examine the selectivity. In addition, during the fluorescent mitochondrial targeted imaging experiments, a control group should be supplemented (-UV-365, then remove noncovalent probes via fixation and washing) to verify that in dark environment the o-nitrobenzyl alcohol reagents does not proceed any undesired labeling toward other endogenous substances inside live cells after extensive washing.

Collectively, based on current development of the PANAC photo-cross-linking chemistry and the progress made in this article, it is not recommended that this manuscript could be published in the Nature Communications, unless all the concerns could be fully resolved.

Reviewer #3 (Remarks to the Author):

Guo and co-workers report a light-induced reaction between primary amines and o-nitrobenzyl alcohols (O-NBA). Based on previous reports that o-nitrobenzyl alcohol could react with amines under UV activation to form indazolones, they found that electron-withdrawing amide group substituent could increase the reaction efficiency of o-nitrobenzyl alcohol. With this O-NBA amide derivatives, they have conducted amine compound and peptide modification, native protein bioconjugation, protein profiling and in vivo protein labeling. The experiments are well executed and are very solid. There are some issues to be addressed:

- 1) the authors spent a page discuss click chemistry, and tried the best to link the chemistry in this manuscript with click chemistry. This reviewer does not see the point to do so and does not agree with such connection.
- 2) there are many reactions in the literature which could rapidly label/modify amine for bioconjugation. Should cite them.
- 3) page 7, the PANAC product stability comparing with maleimide-type conjugates is between apple and orange. There are other amine bioconjugation products which are very stable.
- 4) In Figure 3, only HPLC conversions are provided. The isolated yields of some reactions, if not all, should be given. Based on the HPLC profiles, there are some small peaks. Isolated yields would be more accurate to reflect how well these reactions went.
- 5) For labeling nanobody-HER2, protein aggregation degree should be provided.
- 6) In supporting information, there are shoulder peaks for the PANAC products, such as Figures S10-C/S11-d/S12-d/S34-d, what are they?
- 7) In supporting information, some B-BNA showed two peaks, such as in Figure S15/16/17/18/19/20. Why?

Point-By-Point Reply to the Comments of Reviewer #1

Reviewer #1 (Remarks to the Author):

Chen et.al. reported a photo-initiated cyclization reaction of primary amine and o-nitrobenzyl alcohols. The reaction is actually the conjugation reaction of primary amines with aryl-nitroso generated from photolysis of o-nitrobenzyl alcohols upon UV irradiation. As mentioned in Figure 2A and 2B, this UV-triggered fast cyclization reaction has been reported (J. Org. Chem. 2018, 83, 15493; RSC Adv. 2019, 9, 13249) and applied to the lysine-selective photo-crosslinking to capture protein-protein interaction. The authors screen different o-NBA substrates, then they found that o-NBA amide proceeded fast photolysis as well as the following cyclization with primary amine, which took around 30~40 minutes in total to accomplish. The reaction showed good selectivity to primary amine in the lysine side chain. The reaction seemed to have good compatibility with terminal alkynes and azide, which are functionalities used in CuAAC. Lysine-specific modification of peptides or proteins was demonstrated using this reaction. The authors also showed preliminary applications of this reaction in endogenous kinase profiling and mitochondria-targeted labeling. In general, the work is of good quality and important to the field of bioconjugation, but the following issues still need to be addressed:

1. The importance of CuAAC and SPAAC as click reaction is largely originated from the bioorthogonality of the reactions. However, this reaction can not be bioorthogonal since lysine is abundant in biological systems. In this sense, Figure 1 was not very proper. I would suggest the authors add some introduction to other fast and biocompatible ligation reactions, such as the CBT chemistry. Besides, comparison of this reaction to known amine-involved bioconjugation method with respect to their efficiency in protein labeling is necessary.

Response: We are honestly grateful for the comments from this reviewer. We agree that “this reaction could not be bioorthogonal since lysine is abundant in biological systems”. In fact, we demonstrated the light-induced PANAC reaction as a photoclick reaction in our manuscript.

We have modified the Figure 1 in our revised manuscript as reviewer suggested, and also added some introduction and have cited reference about other fast and biocompatible ligation reactions such as CBT-click reaction (ref. 19-21), tyrosine-click reaction (ref. 22-23), amine bioconjugations (ref. 24-26) and Sulfur(VI) fluoride exchange (SuFEx) chemistry (ref. 7). We have added these citations to the reference part in our revised manuscript.

The comparison of the PANAC photoclick to known amine-involved bioconjugation methods had been discussed in the discussion part. Please see the discussion part, “III. By contrast, among well-established spontaneous primary amine conjugation approaches⁴³, highly amine-reactive electrophiles (e.g. activated ester, aryl sulfonyl fluoride) are probably too prone to non-enzymatic or enzymatic hydrolysis with reduced conjugation efficiency, during in vitro conjugations or during the process into cells as chemical biology probes^{27,28}.” And “IV. Several frequently used amine (or lysine)-reactive electrophiles cross-react with other nucleophiles (e.g. nucleophilic chains of amino acids, O, N-nucleophiles, GSH) giving promiscuous reactivity. We show here that o-NBA derived reactants are able to overcome several aforementioned challenges in complex biological environments, such as background labeling from spontaneous conjugations or cross-reactivity with other nucleophiles, and reduced efficiency from hydrolysis^{27,28}.”

2. The photo-initiation condition may add some advantages to the reaction if spatial control of the reaction can be demonstrated in a well-designed experiment. The importance of temporal control of bioconjugation reactions is not as significant as spatial control, since the other reaction substrate can be introduced to the system with temporal resolution.

Response: We are grateful to reviewer for this suggestion. We have designed experiments to demonstrate the spatial control (also see references: J. Am. Chem. Soc. 2018, 140, 14542-14546; J. Am. Chem. Soc. 2017, 139, 8090-8093; Nat. Chem. 2011, 3, 256-259.) of the photo-induced PANAC reaction (Supplementary Fig. 9-11). We added these results (Supplementary Fig. 9-11) to the revised Supplementary information.

We introduced the o-NBA moiety (compound S40) to the HER2 specific nanobody (sdAb-HLC, Bioconjugate Chem. 2014, 25, 979-988) at C-terminal to produce (sdAb-HLC-o-NBA). The HER2 specific nanobody (sdAb-HLC-o-NBA) was labeled with FITC-NH₂ via PANAC photo-click reaction on the surface of SK-Br-3 cells, providing strong FITC signals (green) on the surface of the cells (Supplementary Fig. 9).

Supplementary Fig. 9 | Schematic illustration of spatial labeling of live cells by PANAC photo-click reaction. a, Incorporation of the o-NBA moiety (compound S40) to the HER2 specific nanobody (sdAb-HLC, Bioconjugate Chem. 2014, 25, 979-988) at C-terminal to produce (sdAb-HLC-o-NBA). b, The structure of the FITC compound. c, The HER2 specific nanobody (sdAb-HLC-o-NBA) labeled with FITC-NH₂ via PANAC photo-click reaction on the surface of SK-Br-3 cells.

Supplementary Fig. 10 | SK-Br-3 Cells were incubated with HER2 specific nanobody (sdAb-HLC-o-NBA) and Hoechst 33324 (blue) at 37 °C for 1 h and 15 min respectively, washed with PBS 3 times, then incubated with FITC-NH₂ (green) and exposed to 365 nm UV light irradiation for 20 min followed by PBS wash 3 times. Cell imaging using confocal fluorescence. Scale bar: 10 μm.

In addition, labeling with spatial control was demonstrated simply by shielding part of the cells from UV-365 nm light irradiation (bottom panel), or with light-activation for 10 min (top panel). Strong FITC signals (green) on the surface of the cells (Supplementary Fig. 11, top panel) indicated the light-induced PANAC reaction of HER2 specific nanobody (sdAb-HLC-o-NBA) with FITC-NH₂. (J. Am. Chem. Soc. 2018, 140, 14542-14546; J. Am. Chem. Soc. 2017, 139, 8090-8093; Nat. Chem. 2011, 3, 256-259.)

Supplementary Fig. 11 | Labeling of HER2 specific nanobody (sdAb-HLC-o-NBA) with FITC-NH₂ on the surface of SK-Br-3 cells via light-induced PANAC reaction. SK-Br-3 Cells were incubated with HER2 specific nanobody (sdAb-HLC-o-NBA) and Hoechst 33324 (blue) at 37 °C for 1 h and 15 min respectively, washed with PBS 3 times. Labeling with spatial control was demonstrated simply by shielding part of the cells from UV-365 nm light irradiation (bottom panel) or with light-activation for 10 min (top panel). After labeling, cells were washed by PBS wash for 3 times. And cell imaging using confocal fluorescence. Scale bar: 10 μm.

3. Original NMR spectra of the products listed in Figure 3 should be provided in supplementary data.

Response: We are grateful to reviewer for this comment. We have added the Original NMR spectra of the products listed in Figure 3 to the revised Supplementary Information on page #87-109.

4. The authors mentioned kinase profiling using the probe containing the o-NBA structure captured much more kinase species than the work using the probe containing the diazirine photo-affinity moiety. It is hard to understand since the carbene species generated from photolysis of diazirine should have much higher chance to form covalent bonds with adjacent groups. The authors may need to comment on this.

Response: The highly reactive carbene intermediate photogenerated from diazirine moiety, which can be readily quenched with water in complex biological environment, or convert to the less reactive species, resulting in short half-life (nanosecond to microsecond range). This event will lead to lower crosslinking efficiency (Angew. Chem. Int. Ed. 2018, 57, 14350). While our o-nitrosobenzaldehyde intermediate has a long half-life (Fig. 2d) and is selective for amine groups, which is not easy to be quenched by water, so o-NBA structure captures more kinase substrates than diazirine photo-affinity moiety.

5. The mitochondria-targeted labeling was not really a spatial control realized by the photo-controllable reaction. More convincing results to show the advantage of this reaction to spatial labeling may need photo-irradiation confined to subcellular level.

Response: We are grateful to reviewer for this suggestion. We have demonstrated the spatial labeling of live cells by PANAC photo-click reaction with pre-tagged o-NBA moiety which was further labeled with FITC-NH₂ via the PANAC photo-click reaction on the surface of SK-Br-3 cells (please see details in response to issue 2 of this reviewer). We further demonstrated the spatial covalent labeling of engineered human histone H2B which localizes at the nucleolus, via PANAC photo-click reaction (Supplementary Fig. 13-15).

We could not get access to the photo-irradiation confined to subcellular level with UV light at this stage. As spatial control of subcellular labeling, we have demonstrated the mitochondria-targeted labeling with spatial control simply by shielding part of the cells from UV-365 nm light irradiation (Supplementary Fig. 12) (J. Am. Chem. Soc. 2018, 140, 14542-14546; J. Am. Chem. Soc. 2017, 139, 8090-8093; Nat. Chem. 2011, 3, 256-259.).

Supplementary Fig. 12 | Confocal images of MDA-MB-486 cells stained with Rho-o-NBA probe (Red channel) and Hoechst 33342 (Blue channel) in different conditions: Rho-o-NBA probe with light irradiation (a), or shielding part of the cells from UV-365 nm light irradiation (b), then fixed with 4% PFA and washed to remove unreacted probe for intended comparison.

We further demonstrated the spatial covalent labeling of engineered human histone H2B which localizes at the nucleolus, via PANAC photo-click reaction (Supplementary Fig. 13). The red fluorescent protein (mKate2) was fused to the C-terminus of H2B as red fluorescent marker. The unnatural amino acid (o-NBAK) was site-specifically incorporated at the site of 175 of mKate2 of the fusion protein (H2B-mKate2-175TAG) via expansion of the genetic code (please see our previous research, ref 41, Chem. 2019, 5, 2955-2968). The fusion protein (H2B-mKate2-175TAG) was expressed in live 293T cells, then the cells were labelled with FITC-NH₂ via PANAC reaction upon light-activation. After labeling, the unreacted FITC-NH₂ was removed by washing, and then imaged with confocal microscopy.

Supplementary Fig. 13 | Schematic illustration of site-specific labeling of H2B-mKate2-175-o-NBAK protein which localizes at the nucleolus, via PANAC photo-click reaction with temporal and spatial control.

After the light-induced PANAC reaction of the FITC-NH₂ with o-NBAK, the FITC signals colocalized with mKate2 signals, suggesting the site-specific labelling of histone H2B (Supplementary Fig. 14).

Supplementary Fig. 14 | Unnatural amino acid o-NBAK was added to the growth medium 1h prior transfection at a final

concentration of 1 mM. HEK293T cells were transfected with pcDNA3.1-H2B-mKate2 K175TAG HEK293T (red channel) and pNEU-hMbPyIRS-4xU6M15 plasmids (Ref: Nucleic Acids Res. 2018, 46, 1-10). After cultured for 24 h, HEK293T cells were fixed, stained with Hoechst 33342 (2 μ g/mL, blue channel) for 15 min, incubated with FITC-NH₂ (20 μ M, green channel) for 15 min and exposed to 365 nm for 20 min. Scale bar: 2 μ m.

As negative controls, there was no signal of FITC found at the nucleolus without the light-activation (Supplementary Fig. 15a). In addition, in the absence of the unnatural amino acid (o-NBAK), there was no signal of mKate2 signal at the nucleolus (Supplementary Fig. 15b), indicating mKate2 signal depending on the site-specific incorporation of the unnatural amino acid (o-NBAK) in H2B-mKate2-175-TAG protein.

Supplementary Fig. 15 | a, with unnatural amino acid o-NBAK was added to the growth medium 1h prior transfection at a final concentration of 1 mM. b, without unnatural amino acid o-NBAK. HEK293T cells were transfected with pcDNA3.1-H2B-mKate2 K175TAG HEK293T (red channel) and pNEU-hMbPyIRS-4xU6M15 plasmids. After cultured for 24 h, HEK293T cells were fixed, stained with Hoechst 33342 (2 μ g/mL, blue channel) for 15 min, incubated with FITC-NH₂ (20 μ M, green channel) for 15 min without light-activation. Scale bar: 2 μ m.

We have added these results (Supplementary Fig. 9-12, 13-15) to the revised Supplementary Information, and discussed the spatial control of the PANAC conjugations in the discussion part of the revised manuscript on the page of #18, with the sentence as: “Alternatively, o-NBA-related handles could be in principle incorporated in native biomolecules such as proteins (Supplementary Fig. 9-12, 13-15)^{18, 41}, followed by exogenous primary-amine probe labeling in spatial and temporal control in vitro or in living systems, which would greatly expand the power of this light-induced PANAC conjugation⁹.”

Point-By-Point Reply to the Comments of Reviewer #2

Reviewer #2 (Remarks to the Author):

Pros.:

1. Via systematic optimization, the developed PANAC reaction now has higher yields for various substrates including drugs targets, peptides and protein substrates via in-vitro chemical modification under different UV irradiation conditions. The selectivity of PANAC reaction toward primary appears to be pretty good. After UV irradiation, the o-nitrosobenzaldehyde amide 3 showed negligible reactivities toward alcohols, phenols, carboxylic acids, and nitrogen-containing heterocycles groups in the presence of primary amine. But there is no data to explain the fate of the intermediate, o-nitrosobenzaldehyde amide 3, toward those endogenous nucleophiles without trapping by primary amines, especially in living systems.

Response: We appreciate the positive comments from this reviewer. We have finished several experiments and provided data to explain the fate of the intermediate, o-nitrosobenzaldehyde amide 3 without trapping by primary amines, in the presence of several potential nucleophiles, such as thiol (Cysteine), nucleophilic side chains of amino acids (also see reference, *Angew. Chem. Int. Ed.* 2012, 51, 6502-6505).

Please see the Supplementary Fig. 28-34 for details. We also have added the note “Supplementary Fig. 28-34” into the revised manuscript on page #7.

2. In this paper, the introduction of amide groups on para-position of o-nitrobenzyl alcohols has increased the reaction efficacy of the well-established PANAC reaction toward solvent exposed primary amine groups, demonstrating a useful photo-cross-linking tool for studying the pharmacological properties of payload molecules via covalent tagging to native primary amines in biological systems. Since the primary amines are key and abundant residues on biomolecules in live cells, it is obvious that this type of PANAC photo-cross-linking reagents hardly show any bioorthogonality.

Response: We are grateful to Reviewer for this comment. In fact, primary-amine-containing molecules are abundant and widespread in synthetic chemistry, biological systems and materials science (*Annu. Rev. Biochem.* 2019, 88, 365-381; *ACS Comb. Sci.* 2015, 17, 600-607). We agree that “this type of PANAC photo-cross-linking reagents hardly show any bioorthogonality”. Indeed, we have demonstrated the light-induced PANAC reaction as a photoclick reaction via diverse applications in our manuscript. Thus, the PANAC photoclick chemistry provides versatile platform for modular conjugation of multiple functional motifs and primary amines—one of the most abundant functional groups as straightforward click handle, conferring expanding the click chemistry toolkit.

Cons.:

1. The novelty of this PANAC reaction is not prominent, it is a continuous study derived from previously research

work reported by the same group in which the o-nitrobenzyl alcohol is specifically incorporated on the desired residues of target proteins (Ref. 31). The PANAC reaction was early reported in 2011 (Chem. Commun., 2011, 47, 3822–3824) for covalent photo-assembly of gold nanoparticles, and has been utilized for bio-material modifications with decent spatiotemporal resolution (e.g. Chem. Mater. 2019, 31, 4710–4719, ACS Macro Lett. 2016, 5, 19–23, Angew. Chem. Int. Ed. 2012, 51, 9181–9184, Adv. Mater. 2016, 28, 2724–2730). The photon quantum efficiency and the reaction rate constant of the second imine formation step have been reported (Ref. 28), but it is not investigated carefully in this article; The choice of light wavelength for triggering the photo-transformation of 2 were also investigated previously (Ref. 29, J. Org. Chem. 2018, 83, 24, 15493-15498). However, when applying to live cell studies as a bio-conjugation chemistry, the authors ignored the photo-toxicity of using 365 nm UV irradiation and potential cytotoxicity caused by extensively blocking of vital primary amines.

Response: We are honestly grateful for the comments from this reviewer. We want to response these comments separately, as follows:

“The novelty of this PANAC reaction is not prominent, it is a continuous study derived from previously research work reported by the same group in which the o-nitrobenzyl alcohol is specifically incorporated on the desired residues of target proteins (Ref. 31). The PANAC reaction was early reported in 2011 (Chem. Commun., 2011, 47, 3822–3824) for covalent photo-assembly of gold nanoparticles.”

Although the previous studies revealed the successful indazolone formations under light activation by other groups (Chem. Commun., 2011, 47, 3822-3824; J. Org. Chem. 2018, 83, 24, 15493-15498) and our group (RSC Adv. 2019, 9, 13249-13253). However, these transformations still suffer from harsh reaction conditions (e.g. long time UV-exposure, 3 - 24 h), and the reaction efficiency (e.g. 3 - 24 h reaction time) and substrate diversity still need to be improved with respect to synthetic applications. Therefore, the generality (i.g. efficiency, accessibility and functionality) of above-mentioned indazolone formations do not meet the criteria of click chemistry, as well as the proximity-enabled protein-protein photo-crosslinking strategy (Chem. 2019, 5, 2955-2968) developed by us previously.

“and has been utilized for bio-material modifications with decent spatiotemporal resolution (e.g. Chem. Mater. 2019, 31, 4710–4719, ACS Macro Lett. 2016, 5, 19–23, Angew. Chem. Int. Ed. 2012, 51, 9181–9184, Adv. Mater. 2016, 28, 2724–2730).”

As this reviewer mentioned, the o-nitrobenzyl alcohol derivative has been utilized for bio-material modifications with decent spatiotemporal resolution (e.g. Chem. Mater. 2019, 31, 4710-4719, ACS Macro Lett. 2016, 5, 19-23, Angew. Chem. Int. Ed. 2012, 51, 9181–9184, Adv. Mater. 2016, 28, 2724-2730). However, in these studies, the photogenerated o-nitroso-benzaldehydes reacted with primary amine forming imine (Chem. Mater. 2019, 31, 4710-4719), with hydrazines forming hydrazone adducts (ACS Macro Lett. 2016, 5, 19-23; Adv. Mater. 2016, 28, 2724-2730), or with hydroxylamine forming oxime (Angew. Chem. Int. Ed. 2012, 51, 9181-9184), respectively. None of the above-mentioned ligation products was indazolone product.

By contrast, in our research, the PANAC photo-click reaction provides the stable cyclization products (indazolones) which are different compared to those products in bio-material modifications. The PANAC product was hydrolytically and thermally stable in buffer conditions (please see, Supplementary Fig. 2). On the other hand, the imine/hydrazone/oxime bonds are reversible in nature (ACS Macro Lett. 2016, 5, 19-23). With respect to these factors, the PANAC photo-click reaction is different in mechanism to those imine/hydrazone/oxime ligations.

“The photon quantum efficiency and the reaction rate constant of the second imine formation step have been reported (Ref. 28), but it is not investigated carefully in this article; The choice of light wavelength for triggering the photo-transformation of 2 were also investigated previously (Ref. 29, *J. Org. Chem.* 2018, 83, 24, 15493-15498).”

In our revised manuscript, we have measured the photon quantum efficiency of the o-nitrosobenzaldehyde photogenerated from o-nitrobenzyl alcohol is 0.52 (please see details in response to issue 3 of this reviewer). In addition, we measured and calculated the reaction rate constant of the PANAC reaction ($k_2 = 92 \text{ M}^{-1} \text{ s}^{-1}$) by LC-MS (please see details in response to issue 4 of this reviewer). We have added the photon quantum efficiency and the reaction rate constant of PANAC reaction in the revised manuscript.

In our previously research, the effectiveness of different light wavelength (254 nm/ 365 nm/ blue light) was investigated for the indazolone formation reaction of o-nitrobenzyl alcohol derivative with primary amine, the light with 365 nm resulted in the best yield than others (*RSC Adv.* 2019, 9, 13249-13253). Thus, in current study, we chose the light wavelength with 365 nm for the PANAC photo-click reaction.

“However, when applying to live cell studies as a bio-conjugation chemistry, the authors ignored the photo-toxicity of using 365 nm UV irradiation and potential cytotoxicity caused by extensively blocking of vital primary amines.”

We are grateful to reviewer for this suggestion. According to the previous research (*Sci. Rep.* 2015, 5, 7724; DOI: 10.1038/srep07724), cells exposed to 365 nm for 60 minutes had no significant effect on cell viability or apoptosis by XTT and FACS experiments. In our studies, we used the low-intensity lamps producing 365 nm UV light, which is a result of the compromise between the spectral properties of o-NBA compounds and biocompatibility of light source. We also conducted a cell viability test with reference to the literature (*Sci. Rep.* 2015, 5, 7724), and found that the results of the experiments (Supplementary Fig. 21, compound only/ compound+365nm light-10 min/ 365nm light-10 min only) had no significant effect on cell viability, probably due to the short time of UV irradiation and low concentration (o-NBA amide 1 μM or Rho-o-NBA, 1 μM) of the probes as we used in our mitochondria-targeted labeling experiments (Figure 6d-e). In addition, we also used the concentration of the probes (o-NBA amide 4 μM or Rho-o-NBA, 4 μM), the results were similar as low concentration of probes (Supplementary Fig. 21).

Supplementary Fig. 21 | MDA-MB-468 cells were incubated with o-NBA amide 1 (1 or 4 μM) / Rho-o-NBA (1 or 4 μM) / DMSO / STS (0.8 μM) for 30 min and were irradiated with 365 nm UV light for 10 min or without UV. After incubation for further 2 days, cell availability was analyzed by CCK8 kit. Staurosporine (STS) -induced apoptosis and cell viability as control. Data are shown as mean \pm SD (n=3).

We have added the Supplementary Fig. 21 into the revised Supplementary Information. And have added the sentence “We wondered if light irradiation conditions might cause sufficient harm to cells. Fortunately, cell availability assay showed that the short time of 365 nm UV irradiation employed in the PANAC reactions as well as in the presence of (1 μM or 4 μM) o-NBA derived probes, did not noticeably affect cell proliferation of MDA-MB-468 cells (Supplementary Fig. 21).” into the revised manuscript on page #13.

We agree with reviewer that “potential cytotoxicity caused by extensively blocking of vital primary amines”. We will investigate more details about the potential cytotoxicity of PANAC photo-click reaction in vivo rather than in cells in our future research, as this reviewer kindly suggested.

2. The most important characteristic of a photo-click reaction is the control of when and where to induce covalent bond formation between a pair of bioorthogonal groups (azide-alkyne, tetrazole-alkene, azide-cyclopropanone), not native groups, e.g. primary amine. However, even the developed PANAC reactions does not appear to have spatial controllability which probably is due to the long life-time (Fig. 2d) and insufficient reactivity of o-nitrosobenzaldehyde amide 3 toward primary amine, resulting in its diffusion and nonrestrictive labeling in the cellular environment. The unsatisfied bimolecular reaction rate was also reflected as the higher yields of desired indazolone 4 after prolonged incubation up to 30 mins (Fig. 2c, 2e and 2f without indication of reaction time). It is essential to demonstrate the spatial covalent labeling for a photo-cross-linking tool.

Response: We are grateful to reviewer for this comment. As the concept of click chemistry was introduced by Sharpless and their colleagues, click chemistry refers to a class of reactions that satisfy certain characteristics, such as modularity, operational simplicity (e.g. be insensitive to oxygen or water), reliable selectivity, high yields, good reaction rate and diversity of the reactants (Angew. Chem. Int. Edit. 2001, 40, 2004-2021; Angew. Chem. Int. Edit. 2013, 52, 5930-5938).

Generally, the click chemistry mainly includes two kinds of reactions in bioconjugations. The first type is the bioorthogonal click reactions with pre-incorporation of exogenous click handles (such as azide-alkyne, tetrazole-alkene, azide-cyclopropanone). The second type is the efficiently chemoselective labeling reactions based on native residues of biomolecules, such as 2-cyanobenzothiazole (CBT) and N-terminal cysteine click reaction (ref. 19-21), tyrosine-click reaction (ref. 22-23), amine bioconjugations (ref. 24-26), sulfur (VI) fluoride exchange reaction (ref. 7), light-induced thiol-alkene/alkyne click reactions (ref 32), and so on. In fact, we have demonstrated the light-induced PANAC reaction as the second type of photodick reaction in our manuscript, but not a bioorthogonal click reaction.

We agree with this reviewer that “It is essential to demonstrate the spatial covalent labeling for a photo-cross-linking tool.” Our previous research (Chem. 2019, 5,2955-2968, Genetically encoded residue-selective photo-crosslinker to capture protein-protein interactions in living cells) has showed the light-induced indazolone formation with spatial control for photo-crosslinking of protein-protein interactions. In addition, we have designed more experiments to demonstrate the spatial covalent labeling of the photo-induced PANAC reaction as follows:

1). Temporal and spatial labeling of live cells on the cell surface, please see the results form (please see the following results in Supplementary Fig. 9 - 11). (also see references: J. Am. Chem. Soc. 2018, 140, 14542-14546; J. Am. Chem. Soc. 2017, 139, 8090-8093; Nat. Chem. 2011, 3, 256-259.). We added these Supplementary Figures and results to the revised Supplementary information.

We introduced the *o*-NBA moiety (compound S40) to the HER2 specific nanobody (sdAb-HLC, Bioconjugate Chem. 2014, 25, 979-988) at C-terminal to produce (sdAb-HLC-*o*-NBA). The HER2 specific nanobody (sdAb-HLC-*o*-NBA) was labeled with FITC-NH₂ via PANAC photo-click reaction on the surface of SK-Br-3 cells, providing strong FITC signals (green) on the surface of the cells (Supplementary Fig. 9).

Supplementary Fig. 9 | Schematic illustration of spatial labeling of live cells by PANAC photo-click reaction. a, Incorporation of the *o*-NBA moiety (compound S40) to the HER2 specific nanobody (sdAb-HLC, Bioconjugate Chem. 2014, 25, 979-988) at C-terminal to produce (sdAb-HLC-*o*-NBA). b, The structure of the FITC compound. c, The HER2 specific nanobody (sdAb-HLC-*o*-NBA) labeled with FITC-NH₂ via PANAC photo-click reaction on the surface of SK-Br-3 cells.

Supplementary Fig. 10 | SK-Br-3 Cells were incubated with HER2 specific nanobody (sdAb-HLC-*o*-NBA) and Hoechst 33324 (blue) at 37 °C for 1 h and 15 min respectively, washed with PBS 3 times, then incubated with FITC-NH₂ (green) and exposed to 365 nm UV light irradiation for 20 min followed by PBS wash 3 times. Cell imaging using confocal fluorescence. Scale bar: 10 μm.

In addition, labeling with spatial control was demonstrated simply by shielding part of the cells from UV-365 nm light irradiation (bottom panel), or with light-activation for 10 min (top panel). Strong FITC signals (green) on the surface of the cells (Supplementary Fig. 11, top panel) indicated the light-induced PANAC reaction of HER2 specific nanobody

(sdAb-HLC-o-NBA) with FITC-NH₂ (J. Am. Chem. Soc. 2018, 140, 14542-14546; J. Am. Chem. Soc. 2017, 139, 8090-8093; Nat. Chem. 2011, 3, 256-259.)

Supplementary Fig. 11 | Labeling of HER2 specific nanobody (sdAb-HLC-o-NBA) with FITC-NH₂ on the surface of SK-Br-3 cells via light-induced PANAC reaction. SK-Br-3 Cells were incubated with HER2 specific nanobody (sdAb-HLC-o-NBA) and Hoechst 33324 (blue) at 37 °C for 1 h and 15 min respectively, washed with PBS 3 times. Labeling with spatial control was demonstrated simply by shielding part of the cells from UV-365 nm light irradiation (bottom panel) or with light-activation for 10 min (top panel). After labeling, cells were washed by PBS wash for 3 times. And cell imaging using confocal fluorescence. Scale bar: 10 μm.

2). Spatial covalent labeling of histone protein H2B at the nucleolus via PANAC photo-click reaction (please see the following results in Supplementary Fig. 13- 15)

We further demonstrated the spatial covalent labeling of engineered human histone H2B which localizes at the nucleolus, via PANAC photo-click reaction (Supplementary Fig. 13). The red fluorescent protein (mKate2) was fused to the C-terminus of H2B as red fluorescent marker. The unnatural amino acid (o-NBAK) was site-specifically incorporated at the site of 175 of mKate2 of the fusion protein (H2B-mKate2-175TAG) via expansion of the genetic code (please see our previous research, ref 41, Chem. 2019, 5, 2955-2968). The fusion protein (H2B-mKate2-175TAG) was expressed in live 293T cells, then the cells were labelled with FITC-NH₂ via PANAC reaction upon light-activation. After labeling, the unreacted FITC-NH₂ was removed by washing, and then imaged with confocal microscopy.

Supplementary Fig. 13 | Schematic illustration of site-specific labeling of H2B-mKate2-175-o-NBAK protein which localizes at the nucleolus, via PANAC photo-click reaction with temporal and spatial control.

After the light-induced PANAC reaction of the FITC-NH₂ with o-NBAK, the FITC signals colocalized with mKate2 signals, suggesting the site-specific labelling of histone H2B (Supplementary Fig. 14).

Supplementary Fig. 14 | Unnatural amino acid o-NBAK was added to the growth medium 1h prior transfection at a final concentration of 1 mM. HEK293T cells were transfected with pcDNA3.1-H2B-mKate2 K175TAG HEK293T (red channel) and pNEU-hMbPyIRS-4xU6M15 plasmids (Ref: Nucleic Acids Res. 2018, 46, 1-10). After cultured for 24 h, HEK293T cells were fixed, stained with Hoechst 33342 (2 μ g/mL, blue channel) for 15 min, incubated with FITC-NH₂ (20 μ M, green channel) for 15 min and exposed to 365 nm for 20 min. Scale bar: 2 μ m.

As negative controls, there was no signal of FITC found at the nucleolus without the light-activation (Supplementary Fig. 15a). In addition, in the absence of the unnatural amino acid (o-NBAK), there was no signal of mKate2 signal at the nucleolus (Supplementary Fig. 15b), indicating mKate2 signal depending on the site-specific incorporation of the unnatural amino acid (o-NBAK) in H2B-mKate2-175-TAG protein.

Supplementary Fig. 15 | a, with unnatural amino acid o-NBAK was added to the growth medium 1h prior transfection at a final concentration of 1 mM. b, without unnatural amino acid o-NBAK. HEK293T cells were transfected with pcDNA3.1-H2B-mKate2 K175TAG HEK293T (red channel) and pNEU-hMbPyIRS-4xU6M15 plasmids. After cultured for 24 h, HEK293T cells were fixed, stained with Hoechst 33342 (2 $\mu\text{g/mL}$, blue channel) for 15 min, incubated with FITC-NH₂ (20 μM , green channel) for 15 min without light-activation. Scale bar: 2 μm .

We have added these results (Supplementary Fig. 9-11, 13-15) to the revised Supplementary Information, and discussed the spatial control of the PANAC conjugations in the discussion part of the revised manuscript with the sentence as: “Alternatively, o-NBA-related handles could be in principle incorporated in native biomolecules such as proteins (Supplementary Fig. 9-12, 13-15)^{18,41}, followed by exogenous primary-amine probe labeling in spatial and temporal control in vitro or in living systems, which would greatly expand the power of this light-induced PANAC conjugation⁹.”

3. The developed PANAC reaction still requires irradiation of 365 nm UV for minimal 7 mins to achieve the moderate to high yield after further 30 mins incubation. The photo-physical properties of o-nitrobenzyl alcohol reagents and the PANAC chemistry were not investigated in this article.

Response: We are honestly grateful for the comment from this reviewer. We have finished several experiments and got data for this issue. The photo-physical properties of o-nitrobenzyl alcohol reagent (o-NBA amide 1) were summarized in (Supplementary Fig. 16- 17), and the photo-physical properties PANAC chemistry please see “the response to this reviewer on issue 4.”

According to the UV-Vis spectrum, o-NBA amide 1 has absorption around 365 nm with $\epsilon = 850 \text{ M}^{-1} \text{ cm}^{-1}$ (Supplementary Fig. 16). In our studies, we chose the 365 nm UV light for light-activation, which is a result of the

compromise between the spectral properties of o-NBA compounds and the biocompatibility of light source. The quantum yield of o-NBA amide 1 under 365 nm excitation to generate o-nitroso benzaldehyde intermediate was 0.52 (Supplementary Fig. 16), which was determined using a ferrioxalate-polyoxometalate-based chemical actinometer (Ref: Jakob Wirz, et al. Photochem. Photobiol. Sci., 2005, 4, 33-42).

Supplementary Fig. 16 | Determining o-NBA amide 1 under 365 nm excitation to generate o-nitroso benzaldehyde quantum yields using a ferrioxalate-polyoxometalate-based chemical actinometer (Ref: Environ. Sci. Technol. 2007, 41, 5433-5438. dx.doi.org/10.1021/ja407867a; J. Am. Chem. Soc. 2013, 135, 16766-16769). (a) Photoinduced formation of POM^- over a period of 30 sec. Data are shown as mean \pm SD (n=3). (b) Reaction of o-NBA amide 1 (20 μM) and methylamine (200 μM) over 120s by UPLC-MS. The reaction is a zero-order reaction with light as a decisive step. Data are shown as mean \pm SD (n=3). (c) Photo-physical properties of o-NBA amide 1 with Ferrioxalate-polyoxometalate-based chemical actinometer as a reference. The absorbance (A_s) of ferrioxalate-polyoxometalate at 365 nm was 2.46 determined by UV-Vis spectrophotometer. The quantum yield (Φ_n) of o-NBA amide 1 under 365 nm excitation to generate o-nitroso benzaldehyde was calculated using the following equation: $\Phi_n = (A_r/A_n) (k_n/k_r) \Phi_r = (\epsilon_r C_r / \epsilon_n C_n) (k_n/k_r) \Phi_r$.

Irradiation of o-NBA in PBS/MeOH at 365 nm induced the formation of new absorption bands at around 282 nm and 307 nm indicating the formation of intermediate which was identified by LC-MS (Supplementary Fig. 17).

We have added the Supplementary Fig. S16-S17 into the revised Supporting Information. And have added the sentence as “with a quantum yield of 0.52 (Supplementary Fig. 16-17)³⁸, the photogenerated intermediate aryl-nitroso 3” into the revised manuscript on page #7.

Supplementary Fig. 17 | (a) UV-Vis spectrum of o-NBA amide 1 (20 μM) in 10 mM PBS/MeOH (1:1, pH = 7.4) irradiated for 0/5 min/10 min/20 min/30 min. (b) photo-physical properties of o-NBA amide 1.

4. The authors claimed that the optimized PANAC reaction is fast and comparable to those of CuAAC, tetrazole photo-click reaction and the tetrazine involved reaction (e.g. *J. Am. Chem. Soc.* 2016, 138, 5978-5983. 660 nm light triggered IEDDA; photo-SPAAC: *J. Am. Chem. Soc.* 2017, 139, 24, 8090-8093). But there was no data of reaction rates specified in the article to support this argument.

Response: We are grateful to reviewer for this comment. We measured and calculated the reaction rate constant of the PANAC photo-click reaction ($k_2 = 92 M^{-1} s^{-1}$) by LC-MS (Supplementary Fig. 18).

Compared to the data from the reference (ref 29, *Chem. Soc. Rev.* 2017, 46, 4895-4950), the reaction rate of light-induced PANAC reaction is comparable to CuAAC (~ 10 - $100 M^{-1} s^{-1}$), tetrazole photo-click reaction (10 - $60 M^{-1} s^{-1}$), the tetrazine involved reactions (1 - $10^4 M^{-1} s^{-1}$), and SPAAC ($\sim 10^2$ - $1 M^{-1} s^{-1}$) or as reviewer mentioned photo-SPAAC ($\sim 40 \pm 2 M^{-1} s^{-1}$, *J. Am. Chem. Soc.* 2017, 139, 24, 8090-8093).

We have added this data to the revised manuscript in the sentence, "Studies on the reaction kinetics (Supplementary Fig. 1) revealed that the second-order rate constant reaches $92 M^{-1} s^{-1}$ which is fast and comparable to those of CuAAC, tetrazole photoclick and certain tetrazines involved reactions¹⁹." in the revised manuscript on page #7.

Supplementary Fig. 18 | Reaction curve of lnC (o-NBA amide 1) over 90s. o-NBA amide 1 (10 μM) and Cbz-Lys-OMe (100 μM) was irradiated under 365 nm in PBS/MeOH. The pseudo-first order reaction rate constant of o-NBA amide 1 (k_{obs}) was 0.0092 s^{-1} . The second-order reaction rate constant $k_2 = k_{\text{obs}}/[\text{Cbz-Lys-OMe}] = 92 \text{ M}^{-1} \text{ s}^{-1}$. Data are shown as mean \pm SD (n=3).

5. The application of PANAC reaction to conjugate peptide and proteins on hydrogel in 3-dimensional space has been previously validated (*Chem. Mater.* 2019, 31, 4710–4719). In-vitro modification of primary amines in either drug molecules (Fig. 3) or peptides (Fig. 4) or proteins (Fig. 5) does not represent the advantage of the PANAC conjugation reaction because there are many other methods to achieve this goal without mediation through photo-irradiation (e.g. SuFEx, *Bioconjugate Chem.* 2017, 28, 1422–1433).

Response: We are grateful to reviewer for these comments, please allow us to express our explanation on this issue.

Firstly, in the research of (*Chem. Mater.* 2019, 31, 4710-4719), the photogenerated o-nitroso-benzaldehydes reacted with primary amine (Boc-Lys) **forming imine as ligation product**, which was referred as **imine ligation**. Indeed, the imine bond or product is reversible in nature (*ACS Macro Lett.* 2016, 5, 19-23), (also see details in response to issue 1 of this reviewer). The indazolone products were never claimed in these bio-material modifications, probably due to the different electronic feature and the position of substituents on the o-nitrobenzyl alcohols backbones. By contrast, in our research, the developed light-Induce reaction provides the stable **cyclization products** (indazolones), thus the light-induced **Primary Amines and o-Nitrobenzyl Alcohols Cyclization** was referred as **PANAC** reaction. The PANAC photo-click reaction provides the hydrolytically and thermally stable cyclization products in buffer conditions (see Supplementary Fig. 2). Therefore, the PANAC photo-click reaction is different in mechanism compared to imine ligation (*Chem. Mater.* 2019, 31, 4710-4719).

Secondly, the o-NBA derived reactants are inert before light activation in our PANAC photo-click reaction, thus could serve as general masked reactants in complex environments. By contrast, among well-established spontaneous primary amine conjugation approaches, highly amine-reactive electrophiles (e.g. activated ester, aryl sulfonyl fluoride) are probably too prone to non-enzymatic or enzymatic hydrolysis with reduced conjugation efficiency, during *in vitro* conjugations or during the process into cells as chemical biology probes (ref 28, *Annu. Rev. Biochem.* 2019, 88, 365-381). In addition, several primary amines-reactive electrophiles and SuFEx click chemistry revealed promiscuous selectivity with other nucleophiles (e.g. nucleophilic chains of amino acids, O, N-nucleophiles, GSH) under complex environments, such as for in

vitro proteins or peptides labeling or in vivo applications. (ref 7, Chem. Soc. Rev. 2019, 48, 4731-4758; ref 27, ACS Chem. Biol. 2017, 12, 1478-1483).

Thirdly, in the research of (Bioconjugate Chem. 2017, 28, 1422–1433) as reviewer mentioned, the dialdehyde click chemistry for amine bioconjugation is in a spontaneous manner without the temporal control. In our research, we have not only showed the applications with primary amines such as drug molecules (Fig. 3) or peptides (Fig. 4) or proteins (Fig. 5) in vitro, but also demonstrated the cell-based and temporally controlled applications, such as profiling of endogenous kinases and organelle-targeted labeling in living systems (Fig. 6). By contrast, the spontaneous manner of the click reactions for native biomolecules remains challenging when applied into living systems, since the reactions would initiate in extracellular environment or during the process approaching cellular targets once certain clickable functional groups are in proximity to each other, thus with increased background labeling from spontaneous conjugations or with reduced efficiency from hydrolysis. The success of the applications via PANAC reaction in living systems mainly based on the intrinsic advantage of temporal control, where the o-NBA derived reactants are inert before light activation, thus serve as general masked reactants in living systems. The advantages of temporal control of photo-click reactions are also discussed in introduction part and discussion part of our manuscript.

Overall, with the unique mechanistic pathway (addition/cyclization/dehydration/tautomerization, Figure 2a) of stable indazolone formations, we have showed that o-NBA derived reactants of our light-induced PANAC reaction are able to overcome several aforementioned challenges in complex biological environments (or in living systems), such as background labeling from spontaneous conjugations or cross-reactivity with other nucleophiles (e.g. nucleophilic chains of amino acids, O, N-nucleophiles, GSH), and reduced efficiency from hydrolysis. The currently developed light-induced PANAC conjugation possesses the unique features and advantages, especially applied into complex biological environments. Please see the discussion part for details in our manuscript.

6. In live cell imaging experiment, there was a significant expansion of the Rho-o-NBA signals after 365 nm irradiation in the cytosolic region of cells in comparing with the Rho-o-NBA channel without the photo-irradiation which was colocalized well with the Rho123 signals (Fig. 6e, first lane, red color). This phenomenon suggested the non-specific cross-linking reaction of the photo-transformed Rho-o-Nitrosobenzaldehyde species toward primary amine residues occurred outside the mitochondria organelles. It is important to add an extra channel via immuno-staining of mitochondrial antigens to identify the specificity of the PANAC photo-cross-linking methods. The fixed cell lysate could also be collected for subsequent in-vitro analysis to examine the selectivity. In addition, during the fluorescent mitochondrial targeted imaging experiments, a control group should be supplemented (-UV-365, then remove noncovalent probes via fixation and washing) to verify that in dark environment the o-nitrobenzyl alcohol reagents does not proceed any undesired labeling toward other endogenous substances inside live cells after extensive washing.

Response: We are grateful to reviewer for this comment. We calculated the mitochondrial area (Fig. 6e, top or bottom panel, red color) and nuclear area of (Fig. 6e, top or bottom panel, blue color) by ImageJ software, respectively. Base on the calculation in Supplementary Fig. 19 (left, data from Fig. 6e), we agree with this reviewer that “there was a significant expansion of the Rho-o-NBA signals (Fig. 6e, bottom panel, red color) in comparing with the Rho-o-NBA channel without the photo-irradiation (Fig. 6e, top panel, red color)”.

Supplementary Fig. 19 | Mitochondrial staining area and Nuclear staining Area of irradiated and unirradiated MDA-MB-468 cells (live cells, or fixed cells from Fig. 6e). The area of Mitochondrial and Nuclear was calculated by ImageJ software, respectively, each calculation ≥ 5 randomly picked cells. Data are shown as mean \pm SD (≥ 5).

We further did the Rho-o-NBA probe labeling *in live cells* with (+ UV-365 nm) or without (– UV-365 nm) light activation (Supplementary Fig. 20), in these experiments, no fixation and no washing procedure was employed. Confocal imaging analysis showed that Rho-o-NBA mitochondria localization merged well with that of Rho 123 (also see calculated data from Supplementary Fig. 19, right, “data from Fig. 19”), which indicated that there was no expansion of the Rho-o-NBA signals with (+ UV-365 nm) or without (– UV-365 nm) light activation *in live cells*.

Supplementary Fig. 20 | Confocal images of MDA-MB-486 cells stained with Rho-o-NBA (Red channel), Rhodamine 123 (Green channel) and Hoechst 33342 (Blue channel) without (top panel) or with (bottom panel) 365nm UV light irradiation. In all of these experiments, no fixation and no washing procedure was employed.

The reason for the “significant expansion of the Rho-o-NBA signals (Fig. 6e, top panel vs bottom panel, red color)” probably resulted from the fixation and extensive washing procedures in the experiment (Fig. 6e, bottom panel, red color). In that experiment, we want to validate that the covalent bond was forming with Rho-o-NBA probe with (+ UV-365 nm) light activation, therefore, we fixed and washed the cells to remove unreacted Rho 123 for intended comparison (Fig. 6e, bottom panel, Rho-o-NBA vs Rho 123 cells staining). However, of note, for regular live cell imaging, due to good cell uptake and subcellular distribution (Fig. 6e, top panel) of Rho-o-NBA probe, it is not necessary to fix cells for imaging or dynamic mitochondrial tracking in living mammalian cells.

Therefore, we have added the note “Signal expansion was observed compared to live cell imaging (Fig. 6e, red color), which probably resulted from the fixation and washing procedures in the experiment, see Supplementary Fig. 19-20 for details.” into the revised manuscript in the figure legends of Fig.6.

“In addition, during the fluorescent mitochondrial targeted imaging experiments, a control group should be supplemented (-UV-365, then remove noncovalent probes via fixation and washing) to verify that in dark environment the o-nitrobenzyl alcohol reagents does not proceed any undesired labeling toward other endogenous substances inside live cells after extensive washing.”

The control group (-UV-365, then remove noncovalent probes via fixation and washing) was displayed in supplementary Fig. S8, which proves the reaction will not proceed in dark environment, as the reviewer suggested.

This result was provided in our last manuscript, as described in the sentence as “Moreover, these reaction-based covalent labeling of the mitochondria is light-dependent event (Supplementary Fig. 8)”.

“Collectively, based on current development of the PANAC photo-cross-linking chemistry and the progress made in this article, it is not recommended that this manuscript could be published in the Nature Communications, unless all the concerns could be fully resolved.”

Response: We have tried our best to resolve all of these concerns as reviewer mentioned, and we hope this reviewer will kindly agree with us. We are honestly grateful for all of these comments and great suggestions from this reviewer.

Point-By-Point Reply to the Comments of Reviewer #3

Reviewer #3 (Remarks to the Author):

Guo and co-workers report a light-induced reaction between primary amines and o-nitrobenzyl alcohols (O-NBA). Based on previous reports that o-nitrobenzyl alcohol could react with amines under UV activation to form indazolones, they found that electron-withdrawing amide group substituent could increase the reaction efficiency of o-nitrobenzyl alcohol. With this O-NBA amide derivatives, they have conducted amine compound and peptide modification, native protein bioconjugation, protein profiling and in vivo protein labeling. The experiments are well executed and are very solid. There are some issues to be addressed:

1. The authors spent a page discuss click chemistry, and tried the best to link the chemistry in this manuscript with click chemistry. This reviewer does not see the point to do so and does not agree with such connection.

Response: We are grateful to reviewer for this comment. We have revised the introduction part and the Figure 1, please see the revised manuscript for details.

2. There are many reactions in the literature which could rapidly label/modify amine for bioconjugation. Should cite them.

Response: We thank Reviewer for this comment. We have added several recently published references about label/modify amines for bioconjugation in reference part (Ref 9, Chem. Biol. 2014, 21, 1075-1101; Ref 24-26, Org. Lett. 2016, 18, 11, 2600-2603; Bioconjugate Chem. 2017, 28, 1422-1433; Nat. Commun. 2019, 10, 142). Several excellent reviews related to amine bioconjugations had already cited in reference part of the manuscript (Ref 28, Annu. Rev. Biochem. 2019, 88, 365-381; Ref 43, Bioconjugate Techniques 3rd Edition, Academic Press, 2013; Ref 44, Chem. Rev. 2015, 115, 2174-2195; Ref 45, J. Am. Chem. Soc. 2019, 141, 2782-2799).

3. Page 7, the PANAC product stability comparing with maleimide-type conjugates is between apple and orange. There are other amine bioconjugation products which are very stable.

Response: We thank Reviewer for this comment. We have switched this sentence “Finally, the PANAC product was hydrolytically and thermally stable in buffer conditions (Supplementary Fig. 2), while certain conjugates (e.g. maleimide-type) are prone to elimination²¹.” to “Finally, the PANAC product was hydrolytically and thermally stable in buffer conditions (Supplementary Fig. 2).”

4. In Figure 3, only HPLC conversions are provided. The isolated yields of some reactions, if not all, should be given. Based on the HPLC profiles, there are some small peaks. Isolated yields would be more accurate to reflect how well these reactions went.

Response: We are grateful to reviewer for this comment. We have obtained several isolated yields of products.

Product 4: > 98% yield (HPLC), 90% yield (isolated). Product 5: 98% yield (HPLC), 94% yield (isolated).

Product 6: > 98% yield (HPLC), 96% yield (isolated). Product 9: > 98% yield (HPLC), 94% yield (isolated).

These results were added to the revised manuscript in Figure 3.

Small peaks on the HPLC profiles are by-products derived from the photogenerated reactive intermediates. We have identified the probable structures of light-induced by-products of *o*-NBA derivatives referred to the literature (Angew. Chem. Int. Ed. 2012, 51, 6502-6505), which were provided in Supplementary Figure 28. We used 4 equivalents of *o*-BNA derivatives to the primary amines in light-induced PANAC reaction, and we also found the excessive intermediates after reaction time. Therefore, these small peaks form the by-products of the photogenerated reactive intermediates probably have no relevance with amine reactants, and they will not affect the reaction yields.

5. For labeling nanobody-HER2, protein aggregation degree should be provided.

Response: For light-induced PANAC covalent labeling of lysines on the nanobody-HER2 with *o*-NBA-alkyne, there was no protein aggregation found. However, in the following CuAAC reaction of alkyne with azide (Fig 5a), with the present of copper ions and additives in the reaction system, it caused partial (~20%) protein aggregation and precipitation, based on the measurement of the protein concentration of the nanobody-HER2. We added this data in figure legends of Fig.5, in the sentence as “in CuAAC, partial (~20%) modified nanobody-HER2 protein aggregation and precipitation.”

On the other hand, the GE-SEC column analysis showed that the modified nanobody-HER2 obtained after desalting had no dimerization and multimerization according to the retention time of the monomer nanobody-HER2 (13.7 kDa) and the reference protein α -Chymotrypsinogen A (25.6 kDa)

6. In supporting information, there are shoulder peaks for the PANAC products, such as Figures S10-C/S11-d/S12-d/S34-d, what are they?

Response: We thank Reviewer for this comment. The PANAC products (indzalones) contain multiple functional groups showing complex acidity and basicity. The acidity of the mobile phase with additives (0.01% formic acid in H₂O, and 0.01% formic acid in CH₃CN) for UPLC are not enough to make these samples completely free or ionized, so they appear as shoulders in the UPLC-MS spectrum analysis.

Therefore, we reduced sample loading, meanwhile, increased the acidity of the mobile phase (0.05% TFA in H₂O and CH₃CN) and changed flow method of partial samples to obtain a symmetrical peak under the same reaction conditions. We

have added these new UPLC traces to the Supplementary Figure 23/24/25/55 part, in the revised Supplementary Information.

7. In supporting information, some o-BNA showed two peaks, such as in Figure S15/16/17/18/19/20. Why?

Response: We are grateful to reviewer for this comment. In the light-induced PANAC reaction, the starting material o-BNA derivatives are excessive (4 equivalent to the primary amines) in the reactions of Figure 3. In Figure S15/16/17/18/19/20, some o-BNA showed two peaks, probably due to the excessive sample loading which exceed the capacity of the column on UPLC-MS analysis.

After reducing the sample loading on UPLC-MS analysis, we obtained a symmetrical single peak in Figure S15/16/17/18/19/20. We have added these new UPLC traces to the Supplementary Figure 35/36/37/38/39/40 part, in the revised Supplementary Information.

REVIEWER COMMENTS

Reviewer #1 (Remarks to the Author):

The author has addressed most of my comments. The revised manuscript was of improved quality and I would recommend it for publication in Nat. Commun. One minor concern is about the relative amount of the two substrates shown in Fig. 2f and Figure 3. Figure 2f showed the use of excess amount of lysine derivative and primary amine substrate, Figure 3 used amine at 0.5mM and o-NBA derivatives at 2 mM. Does that mean that the yields were calculated based on different substrates but all could be very high?

Reviewer #2 (Remarks to the Author):

Based on the revised data, the PANAC reactions displayed decent selectivity to primary amines on the side chain of lysine residues over others. The bimolecular reaction rate constant of PANAC photo-click reaction is also decent and comparable to those of CuAAC, tetrazole photo-click and certain tetrazines in their early generations, but is not enough for highly spatial-resolved applications. The authors also showed the low photo-toxicity of o-NBA reagents by using 365 nm UV irradiation at very low dose (1 or 4 μ M). With more supplementary data provided, the PANAC photo-reaction investigated in this article was proved to be a photo-crosslinking method, which has been utilized recently. This research is based on previously discovered photo-reaction with slight modification of the substituents to improve the performance for bio-conjugation. Innovation of this study is not enough for publication in Nature Communication, but the results are sufficient to support this chemistry as a photo-conjugation method. However, there are still several issues that must be addressed before further consideration.

Although there were reports in the previous research that exposure to 365 nm for 60 minutes had no significant effect on cell viability or leading to apoptosis. The authors should choose 20 minutes or longer time to test the photo-toxicity instead of 10 min, because the irradiation time was more than 10 minutes in the live cell surface labeling or mitochondria tracking experiments, e.g. in the supplementary Fig. 10 and Fig. 13. What's the power density (in unit of mW cm⁻¹) of the UV lamps used for live cell studies as well as for the other experiments? Since the UV-Vis spectra of o-NBA amide 1 showed very weak absorption at 365 nm ($\epsilon_{365} = 850 \text{ M}^{-1} \text{ cm}^{-1}$, Supplementary Fig. 17), the authors should also show the emission spectra of the 365 nm light source used in this study.

As for photo-conjugation toward small molecule conjugations, drug modifications and peptide cyclizations, the PANAC photo-reaction were triggered by 365 nm UV light and then usually require incubation at 25 °C for 30 min to complete the conjugation, because the secondary reaction rate is not fast enough. But for live cell studies in supplementary Fig. 10 or Fig. 11, there was no information on whether it is necessary to incubate for certain time after irradiation for a successful completion of the PANAC bio-conjugation. If the important incubation was no longer needed for this reaction in live cell application, why is it also necessary for small molecule and peptides? Please indicate.

Upon irradiation of 365 nm UV for 10 min with partial shielding, the temporal and spatial labeling performance of PANAC reactions on live cells surface was demonstrated in supplementary Fig. 11. However, it is better to have cells presented cross the dotted line where the spatial resolution on single cell could be explained on the edge of exposure area (e.g. J. Am. Chem. Soc. 2018, 140, 14542-14546; J. Am. Chem. Soc. 2017,139,8090-8093). Currently, the images in supplementary Fig. 11 are not convincing. Why don't show those images in the maintext since it is a characteristic nature of a photo-click reaction?

For the experimental investigation and calculation of the reaction rate constant of PANAC, the bimolecular rate constant k_2 value derived from a single concentration of Cbz-Lys-OMe was not rigorous enough for a photo-click type reaction. The authors should plot various concentration of Cbz-Lys-OMe versus observed apparent first-order rate to derive the k_2 . The deviation of the linear fitting curve, especially in the initial stage of the photo-conversion, was not negligible. Please add more data points and explain.

In the live cell study via mitochondria targeting Rho-o-NBA, confocal images from the Rho-o-NBA channels did not show obvious changes under conditions with (lower row) or without photo-triggering (upper row) when cells are still alive, indicating the rhodamine probe most likely was enriched and well-confined in the mitochondria organelles, reflecting its nature for probing mitochondria. However, these phenomena do not represent the reactivity and specificity of the PANAC reaction for blocking the primary amine inside mitochondria. The authors claim that the significant expansion was probably caused by fixation and extensive washing. On the contrary, the fixation procedure disrupts the membrane integrity of mitochondria and washing may cause the diffusion of the released intermediate Rho-o-Nitrosobenzaldehyde, leading to non-specific conjugations toward primary amines outside the mitochondria with expanded fluorescence signal readout (Fig. 6, Supplementary Fig. 19). This is probably the evidence of incomplete reaction of Rho-o-Nitrosobenzaldehyde intermediate. Although I advised the authors to clarify the point of specificity via an immune-staining of mitochondria via fluorescent anti-body because the cells were fixed, it seems there was no direct answer to this important question. Alternatively, longer incubation time after photo-irradiation for live cell experiments might solve this problem!

Reviewer #3 (Remarks to the Author):

The previous comments raised have been addressed.

Point-By-Point Reply to the Comments of Reviewer #1 (from the second review round)

Reviewer #1 (Remarks to the Author):

The author has addressed most of my comments. The revised manuscript was of improved quality and I would recommend it for publication in Nat. Commun. One minor concern is about the relative amount of the two substrates shown in Fig. 2f and Figure 3. Figure 2f showed the use of excess amount of lysine derivative and primary amine substrate, Figure 3 used amine at 0.5mM and o-NBA derivatives at 2 mM. Does that mean that the yields were calculated based on different substrates but all could be very high?

Response: We appreciate the positive comments from this reviewer.

In fact, we always used the excess amount of o-NBA derivatives in Figure 2f and Figure 3. The yields are all very high in Figure 2f and Figure 3.

As showed in Figure 2f, we used the excess amount of o-NBA derivative (o-NBA amide). In most cases of Figure 2f, the ratio of (o-NBA derivative, 1) to (primary amine substrate, 2) is 4. (please see Figure. 2f, entry 1-4 and entry 6-7, $1 : 2 = 4$). In other examples of Figure 2f, the ratio of (o-NBA derivative, 1) to (primary amine substrate, 2) is ≥ 2 . (please see Figure. 2f, entry 5 and entry 8-10, $1 : 2 \geq 2$).

In Figure 3, we used o-NBA derivatives at 2 mM and primary amines at 0.5 mM, thus the ratio of (o-NBA derivatives) to (primary amines) also is 4.

Point-By-Point Reply to the Comments of Reviewer #2 (from the second review round)

Reviewer #2 (Remarks to the Author):

Based on the revised data, the PANAC reactions displayed decent selectivity to primary amines on the side chain of lysine residues over others. The bimolecular reaction rate constant of PANAC photo-click reaction is also decent and comparable to those of CuAAC, tetrazole photo-click and certain tetrazines in their early generations, but is not enough for highly spatial-resolved applications. The authors also showed the low photo-toxicity of o-NBA reagents by using 365 nm UV irradiation at very low dose (1 or 4 μM). With more supplementary data provided, the PANAC photo-reaction investigated in this article was proved to be a photo-crosslinking method, which has been utilized recently. This research is based on previously discovered photo-reaction with slight modification of the substituents to improve the performance for bio-conjugation. Innovation of this study is not enough for publication in Nature Communication, but the results are sufficient to support this chemistry as a photo-conjugation method. However, there are still several issues that must be addressed before further consideration.

1. Although there were reports in the previous research that exposure to 365 nm for 60 minutes had no significant effect on cell viability or leading to apoptosis. The authors should choose 20 minutes or longer time to test the photo-toxicity instead of 10 min, because the irradiation time was more than 10 minutes in the live cell surface labeling or mitochondria tracking experiments, e.g. in the supplementary Fig. 10 and Fig. 13.

Response: We are honestly grateful for the comments from this reviewer. The irradiation time was 20 minutes in the supplementary Fig. 10 and Fig. 13.

Therefore, as the reviewer suggested, we extended the longer irradiation time (20 minutes and 30minutes) to test the photo-toxicity on cell viability. Based on the results from cell availability analyzed by CCK8 kit, there was no significant effect on cell viability. These results were added to the supplementary information, please see the revised Supplementary Fig. 21b.

2. *What's the power density (in unit of mW cm^{-2}) of the UV lamps used for live cell studies as well as for the other experiments?* Since the UV-Vis spectra of o-NBA amide 1 showed very weak absorption at 365 nm ($\epsilon_{365} = 850 \text{ M}^{-1} \text{ cm}^{-1}$, Supplementary Fig. 17), the authors should also show the emission spectra of the 365 nm light source used in this study.

Response: We appreciate the reviewer for this comment. The irradiation intensity of the UV lamp is $\sim 30 \text{ mW cm}^{-2}$ for the live cell studies as well as for the other experiments.

In fact, the o-nitrobenzyl alcohol (o-NBA) compounds always show weak absorption at 365 nm, and the

365 nm light source was always used for the activation of the *o*-nitrobenzyl alcohols. (please see references: Photochemical Reaction Mechanisms of 2-Nitrobenzyl Compounds: Methyl Ethers and Caged ATP, *J. Am. Chem. Soc.* 2004, 126, 4581-4595; Photochemical reaction mechanisms of 2-nitrobenzyl compounds: 2-nitrobenzyl alcohols form 2-nitroso hydrates by dual proton transfer, *Photochem. Photobiol. Sci.*, 2005, 4, 33-42). In these excellent papers, Jakob Wirz and colleagues measured the absorption of *o*-nitrobenzyl alcohol compounds, and also used 365 nm light source for the activation of *o*-nitrobenzyl alcohols. In addition, Mark J. Kurth and colleagues also found 365 nm light source is best for activation of the *o*-nitrobenzyl alcohols (*o*-NBA), and used the 365 nm light source for photochemical preparation of indazolones (please see reference: *J. Org. Chem.* 2018, 83, 15493-15498).

Last time, we had measured and showed the absorption spectra of *o*-NBA amide 1 in Supplementary Fig. 17, which is consistent with the results of Jakob Wirz and colleagues' reports. We have checked different light wavelength (254 nm/ 365 nm/ 420 nm) for the PANAC photo-click reactions, the light with 365 nm resulted in the best yield than others (also see our previously research, *RSC Adv.* 2019, 9, 13249-13253). More importantly, the 365 nm light source would potentially be less damage to biological samples (ig. proteins, cells), compared to the UV-254/305 nm light. Thus, in current study, we chose the light wavelength with 365 nm for the PANAC photo-click reaction.

Based on above-mentioned results from other groups and our experiments, even though *o*-NBA amide 1 showed weak absorption at 365 nm (Supplementary Fig. 17), the power density from 365 nm light source is enough, and probably more suitable for activation of the *o*-nitrobenzyl alcohol derivatives (*o*-NBA amides), when applied the PANAC photo-click reaction in vitro for biological samples or in biological environment.

As the reviewer suggested, we had checked the emission spectra of the 365 nm light source used in this study. The spectrum of lighting systems for this PANAC photo-click reactions ranges from 350 to 390 nm. The Supplier of hand-held UV lamp: ZF-7A, portable UV lamp, Shanghai Gucun Electron Optic Instrument Factory, China.

As an additional control experiment, we also used the Kessil PR160, UVA LED 370nm lamp (Kessil PR160-370, the spectrum of lighting systems ranges from 355 to 400 nm) as light source, this 370nm light source also provided very high to excellent yields for the PANAC photo-click reactions, at identical conditions.

3. As for photo-conjugation toward small molecule conjugations, drug modifications and peptide cyclizations, the PANAC photo-reaction were triggered by 365 nm UV light and then usually require incubation at 25 °C for 30 min to complete the conjugation, because the secondary reaction rate is not fast enough. But for live cell studies in supplementary Fig. 10 or Fig. 11, there was no information on whether it is necessary to incubate for certain time after irradiation for a successful completion of the PANAC bio-conjugation. If the important incubation was no longer needed for this reaction in live cell application, why is it also necessary for small molecule and peptides? Please indicate.

Response: We are grateful for the comments from this reviewer. As we showed in Figure 2e, the PANAC photo-reaction provides high yields (e.g. about 80% yield, 7 min) after hundreds seconds of light activation (Figure 2e, see the curve: After hv, r.t. 0 min), and a further incubation (Figure 2e, see the curve: After hv, r.t. 30 min) of the reaction mixture is better to give increasing yields (up to 98% yield).

With these results in hand, to get even higher yields of the PANAC photo-reactions, we chose a further incubation for the small molecule conjugations, drug modifications and peptide cyclizations. If without the further incubation for some small molecule conjugations, the yields will be a little (about 20%) lower as provided in our pilot experiments. This is why the PANAC photo-reactions were triggered by 365 nm UV light, and then usually require incubation at 25 °C for 30 min for the experiments of small molecule and peptides. Overall, only to get even higher yields, the further incubation was used for the conjugation experiments of small molecule and peptides.

In our pilot experiments of labeling of live cells on the cell surface, we found the signal intensity is very strong compared to background, thus the signal intensity is enough for cell imaging using confocal fluorescence. This is why there was no further incubation after irradiation for the live cell studies in supplementary Fig. 10 or Fig. 11.

4. Upon irradiation of 365 nm UV for 10 min with partial shielding, the temporal and spatial labeling performance of PANAC reactions on live cells surface was demonstrated in supplementary Fig. 11. However, it is better to have cells presented cross the dotted line where the spatial resolution on single cell could be explained on the edge of exposure area (e.g. J. Am. Chem. Soc. 2018, 140, 14542-14546; J. Am. Chem. Soc. 2017, 139, 8090-8093). Currently, the images in supplementary *Fig. 11 are not convincing. Why don't show those images in the maintext since it is a characteristic nature of a photo-click reaction?*

Response: We are honestly grateful for the comments from this reviewer. We had already showed the spatial labeling of biomolecules via the PANAC photo-click reaction under biological environment such as: **1)** Temporal and spatial labeling of live cells on the cell surface (please see the results in Supplementary Fig. 10); **2)** Spatial covalent labeling of histone protein H2B at the nucleolus via PANAC photo-click reaction (please see the results in Supplementary Fig. 13- 15).

These results already demonstrated the spatial labeling of biomolecules on the scale of single cell or subcellular level, via the PANAC photo-click reaction under biological environment as the reviewers suggested in the last review comments. We could not get access to the spatial resolution on single cell scale with the photo-irradiation cross the single cell for the supplementary Fig. 11, at this stage.

According to the length limits and figure limits of this article content, we chose those images as supplementary information. In addition, we had discussed these results as spatial labeling of biomolecules via the PANAC photo-click reaction under biological environment, in the discussion part of our manuscript.

5. For the experimental investigation and calculation of the reaction rate constant of PANAC, the bimolecular rate constant k_2 value derived from a single concentration of Cbz-Lys-OMe was not rigorous enough for a photo-click type reaction. The authors should plot various concentration of Cbz-Lys-OMe versus observed apparent first-order rate to derive the k_2 . The deviation of the linear fitting curve, especially in the initial stage of the photo-conversion, was not negligible. Please add more data points and explain.

Response: We are honestly grateful for the comments from this reviewer. We have got results of additional four concentrations (50 μ M/ 150 μ M/ 200 μ M/ 250 μ M) of Cbz-Lys-OMe for investigation and calculation of the reaction rate constant of PANAC reaction. Thus, in the revised the reaction rate constant of PANAC reaction including five concentrations of Cbz-Lys-OMe (50 μ M/ 100 μ M/ 150 μ M/ 200 μ M/ 250 μ M) versus observed apparent first-order rate.

In addition, we also have added additional data points of different reaction times such 40s/ 50s/ 70s/ 80s. Therefore, the revised curves have nine different reaction time points (10s/ 20s/ 30s/ 40s/ 50s/ 60s/ 70s/ 80s/ 90s) to measured and calculated the reaction rate constant of the PANAC photo-click reaction with each concentration of Cbz-Lys-OMe, via LC-MS (Supplementary Fig. 18). In these conditions, the deviation of the linear fitting curves were revised and improved.

Based on these additional results and improvement, we have plotted various concentration of Cbz-Lys-OMe versus observed apparent first-order rate to derive the K_2 . Thus, the revised reaction rate constant of the PANAC photo-click reaction is ($k_2 = 87.4 \text{ M}^{-1} \text{ s}^{-1}$) by LC-MS, please see the revised Supplementary Fig. 18.

6. In the live cell study via mitochondria targeting Rho-o-NBA, confocal images from the Rho-o-NBA channels did not show obvious changes under conditions with (lower row) or without photo-triggering (upper row) when cells are still alive, indicating the rhodamine probe most likely was enriched and well-confined in the mitochondria organelles, reflecting its nature for probing mitochondria. However, these phenomena do not represent the reactivity and specificity of the PANAC reaction for blocking the primary amine inside mitochondria. The authors claim that the significant expansion was probably caused by fixation and extensive washing. On the contrary, the fixation procedure disrupts the membrane integrity of mitochondria and washing may cause the diffusion of the released intermediate Rho-o-Nitrosobenzaldehyde, leading to non-specific conjugations toward primary amines outside the mitochondria with expanded fluorescence signal readout (Fig. 6, Supplementary Fig. 19). This is probably the evidence of incomplete reaction of Rho-o-Nitrosobenzaldehyde intermediate. Although I advised the authors to clarify the point of specificity via an immune-staining of mitochondria via fluorescent anti-body because the cells were fixed, it seems there was no direct answer to this important question. Alternatively, longer incubation time after photo-irradiation for live cell experiments might solve this problem!

Response: We are very grateful to the reviewer for these great comments and suggestion.

Since the data in Figure 6e were used to demonstrate the light-induced mitochondria-targeted labeling in **LIVE** cells, an additional experiment in live cells transfected by an EGFP reporter plasmid with specific mitochondria localization (mito-EGFP, see reference: *Neurosci Bull.* 2017, 33(6), 685–694) was performed (see Figure 6e, bottom panel). The co-localization of red fluorescence (from our Rho-o-NBA probe) and green fluorescence (from the mito-EGFP reporter) was observed with (bottom panels) or without light activation (Supplementary Fig. 20 top panels), indicating the good enrichment and confine of Rho-o-NBA probe in the mitochondria organelles in live cells (no fixation and no washing procedure was used).

Using this experimental strategy, the potential effects of fixation, extensive washing, dye diffusion, the unspecific results of immunofluorescence staining, and extended incubation time for the PANAC reaction on the **FIXED** cells could be avoided.

The sole purpose in fixing and washing the cells (former Figure 6e, bottom panel), is to validate that the covalent bond was forming with Rho-o-NBA probe upon (+UV-365 nm) light activation. In other words, the purpose in destructing the live cells (to fix cells) is to verify whether the covalent bond was formed or not, with Rho-o-NBA probe and mitochondria organelle upon light activation(+UV-365 nm), rather than the purpose in live cell imaging.

Overall, the new data added in Figure 6e (bottom panel) is the direct evidence to show the well co-localization of Rho-o-NBA probe (red) and Mito-EGFP (Green), thus, reflecting the specificity of the PANAC photo-cross-linking of Rho-o-NBA probe with mitochondria, as one of application of our PANAC photo-click reaction in living cell imaging.

The results of the fixed cell imaging (former Figure 6e, bottom panel) had already existed in (Supplementary Fig. 8, top panel), which served as supporting evidence for the covalent bond formation between Rho-o-NBA probe and mitochondria via PANAC photo-click reaction.

Point-By-Point Reply to the Comments of Reviewer #3 (from the second review round)

Reviewer #3 (Remarks to the Author):

The previous comments raised have been addressed.

Response: We appreciate this reviewer for the comment.

REVIEWERS' COMMENTS

Reviewer #2 (Remarks to the Author):

After reviewing the revised manuscript, the authors answered and explained most of my queries better. The doubtful point now is why the authors are always reluctant to put the imaging data set for the spatial-resolved labelling (Supplementary Fig. 11 and 12) via the PANAC photo-click reaction in the maintext. It is a characteristic property of a photo-click reaction as I mentioned for several times. Is it really because of the limitation of the number of figures in an article? If this is the reason, I strongly suggest the authors could replace some of less important data in figures, e.g. fig. 3 or 5f or 6e. In addition, the authors did not provide any picture and detailed description to explain how to achieve the partial-shielded exposure, making it is very difficult for readers and researchers to reproduce this PANAC photo-click reaction in implementing spatiotemporal utilization. Therefore, I strongly do not suggest that the manuscript can be published if the authors does not solve this problem very well.

Point-By-Point Reply to the Comments of Reviewer #2 (from the third review round)

Reviewer #2 (Remarks to the Author):

After reviewing the revised manuscript, the authors answered and explained most of my queries better. The doubtful point now is why the authors are always reluctant to put the imaging data set for the spatial-resolved labelling (Supplementary Fig. 11 and 12) via the PANAC photo-click reaction in the main text. It is a characteristic property of a photo-click reaction as I mentioned for several times. Is it really because of the limitation of the number of figures in an article? If this is the reason, I strongly suggest the authors could replace some of less important data in figures, e.g. fig. 3 or 5f or 6e. In addition, the authors did not provide any picture and detailed description to explain how to achieve the partial-shielded exposure, making it is very difficult for readers and researchers to reproduce this PANAC photo-click reaction in implementing spatiotemporal utilization. Therefore, I strongly do not suggest that the manuscript can be published if the authors does not solve this problem very well.

Response: We are honestly grateful for the comments from this reviewer.

As the review suggested, we have added the spatial-resolved labelling (Supplementary Fig. 11 and 12) via the PANAC photo-click reaction as Figure 7 in the main text.

We also have added the results of this figure to the “Light-induced PANAC bioconjugations for native biomolecules in living systems” part in the main text as this sentence: “*On the other hand*, as spatial control for live cell labeling (Supplementary Fig. 9-12, 13-15), we demonstrated the cell surface labeling with the HER2 specific nanobody (Figure 7a) and mitochondria-targeted labeling (Figure 7b) via PANAC photo-click reaction, simply by shielding part of the cells from light-(bottom panel), or with UV-365 nm light-activation (top panel).”

In addition, we have provided the experiment details for the partial-shielded exposure and spatial control of live cell labeling in supplementary information as the review suggested. These experiment details can be found in the section 5.14 and 5.16 of Supplementary Information, respectively.